# Ultrafast charge transfer in mixed-dimensional $WO_{3-x}$ nanowire/$WSe_2$ heterostructures for attomolar-level molecular sensing

Qian Lv [1,10], Junyang Tan [2,10], Zhijie Wang [2,10], Peng Gu [3,4,10], Haiyun Liu [3], Lingxiao Yu [1], Yinping Wei [2], Lin Gan [2], Bilu Liu [2] ✉, Jia Li [2,5] ✉, Feiyu Kang [2,5,6], Hui-Ming Cheng [2,7], Qihua Xiong [3,4,8,9] ✉ & Ruitao Lv [1,6] ✉

Developing efficient noble-metal-free surface-enhanced Raman scattering (SERS) substrates and unveiling the underlying mechanism is crucial for ultrasensitive molecular sensing. Herein, we report a facile synthesis of mixed-dimensional heterostructures via oxygen plasma treatments of two-dimensional (2D) materials. As a proof-of-concept, 1D/2D $WO_{3-x}$/$WSe_2$ heterostructures with good controllability and reproducibility are synthesized, in which 1D $WO_{3-x}$ nanowire patterns are laterally arranged along the three-fold symmetric directions of 2D $WSe_2$. The $WO_{3-x}$/$WSe_2$ heterostructures exhibited high molecular sensitivity, with a limit of detection of $5 \times 10^{-18}$ M and an enhancement factor of $5.0 \times 10^{11}$ for methylene blue molecules, even in mixed solutions. We associate the ultrasensitive performance to the efficient charge transfer induced by the unique structures of 1D $WO_{3-x}$ nanowires and the effective interlayer coupling of the heterostructures. We observed a charge transfer timescale of around 1.0 picosecond via ultrafast transient spectroscopy. Our work provides an alternative strategy for the synthesis of 1D nanostructures from 2D materials and offers insights on the role of ultrafast charge transfer mechanisms in plasmon-free SERS-based molecular sensing.

Ultrasensitive molecular sensing is of great significance in many fields including homeland security, energy and environmental science, clinical diagnosis[1–4]. Compared with the traditional techniques (e.g., electrochemical sensing, mass spectrometry, polymerase chain reaction, etc.), surface-enhanced Raman scattering (SERS) is a fast and non-destructive technique with both high selectivity and sensitivity for the detection of trace amount of molecules or even a single molecule[5]. Noble-metal-based nanomaterials and nanostructures (e.g., Au, Ag),

[1]State Key Laboratory of New Ceramics and Fine Processing, School of Materials Science and Engineering, Tsinghua University, Beijing 100084, China. [2]Shenzhen Geim Graphene Center, Tsinghua-Berkeley Shenzhen Institute and Institute of Materials Research, Shenzhen International Graduate School, Tsinghua University, Shenzhen 518055, China. [3]Beijing Academy of Quantum Information Sciences, Beijing 100193, China. [4]State Key Laboratory of Low-Dimensional Quantum Physics and Department of Physics, Tsinghua University, Beijing 100084, China. [5]Guangdong Provincial Key Laboratory of Thermal Management Engineering and Materials, Tsinghua Shenzhen International Graduate School, Tsinghua University, Shenzhen 518055, China. [6]Key Laboratory of Advanced Materials (MOE), School of Materials Science and Engineering, Tsinghua University, Beijing 100084, China. [7]Shenyang National Laboratory for Materials Science, Institute of Metal Research, Chinese Academy of Sciences, Shenyang 110016, China. [8]Frontier Science Center for Quantum Information, Beijing 100084, China. [9]Collaborative Innovation Center of Quantum Matter, Beijing, China. [10]These authors contributed equally: Qian Lv, Junyang Tan, Zhijie Wang, Peng Gu. ✉e-mail: bilu.liu@sz.tsinghua.edu.cn; li.jia@sz.tsinghua.edu.cn; qihua_xiong@tsinghua.edu.cn; lvruitao@tsinghua.edu.cn

based on the electromagnetic mechanism (EM), offer a wide range of possibilities[6], nonetheless the limited resources and low surface uniformity constrain their widespread applications. In the past decade, two-dimensional (2D) materials (e.g., graphene, transition metal dichalcogenides (TMDCs)) have been intensively pursued as promising noble-metal-free SERS substrates due to their atomically flat surface, good chemical stability, and tunable electronic structures[7–10]. The SERS effect of 2D materials is usually attributed to the chemical mechanism (CM)[11], induced from the charge transfer between the substrates and the probe molecules. However, the limits of detection (LODs) of noble-metal-free 2D materials in previous reports are usually in the range of $10^{-9}$–$10^{-15}$ M[12–18]. Integrating 2D TMDCs with other nanostructures to construct mixed-dimensional heterostructures, such as 2D-1D, might further boost the SERS performance due to the unique electronic structures and physicochemical properties towards enhanced functionalities[19]. In addition, the interactions between such hybrid materials and the analytes often remain elusive and are imperative to be unveiled by time-resolved spectroscopy for the ultrasensitive molecular sensing[20–22].

At present, the synthesis strategies inevitably hinder the practical applications of TMDCs-based heterostructures in integrated nanoelectronics or sensors[23,24]. For example, aligned transfer can lead to the stacking of 2D or 1D materials into vertical heterostructures with well-designed sequences and desired angles, but it is hard to be scaled up[25]. Chemical vapor deposition (CVD) is a feasible route to synthesize large-area TMDC-based heterostructures, but the controllability and repeatability are severely influenced by many factors[26,27]. In comparison, post treatment is a direct method to prepare heterostructures by selectively converting the target area of TMDCs into foreign phases or foreign components[28]. Lin et al.[29] directly synthesized MoX (M = Mo or W, X = S or Se) nanowires by steering a focused electron beam, which were connected to the TMDCs monolayer forming 1D-2D heterostructures. However, complex and precise procedures, as well as rigorous conditions with poor scalability, are usually inevitable. Thus, developing a facile and efficient route for the synthesis of mixed-dimensional heterostructures is crucial but still very challenging.

Herein, we propose an oxygen plasma treatment strategy to synthesize mixed-dimensional heterostructures. As a proof-of-concept, 1D/2D $WO_{3-x}$/$WSe_2$ heterostructures are synthesized by selectively converting the top $WSe_2$ layer to $WO_{3-x}$ nanowires. The 1D $WO_{3-x}$ nanowires are laterally arranged along the three-fold symmetric directions of $WSe_2$. The $WO_{3-x}$/$WSe_2$ heterostructures demonstrate an ultrasensitive SERS effect, which can reach a low LOD of $5 \times 10^{-18}$ M and a high enhancement factor (EF) of $5.0 \times 10^{11}$ for probing methylene blue (MB) molecules. The ultrasensitive SERS capability of heterostructures can be attributed to the efficient charge transfer induced by the unique structures of $WO_{3-x}$ nanowires and the effective interlayer coupling of the heterostructures. Furthermore, we unveil that the charge transfer timescale is around 1.0 ps by the transient spectroscopy, illustrating the ultrafast charge transfer process between the heterostructures and the probe molecules.

## Results

### Structural transformation induced by oxygen plasma treatment

Monolayer and few-layer $WSe_2$ flakes were synthesized with high (8:1) and low (8:3) weight ratios of $WO_3$ and NaCl, respectively, via an atmospheric-pressure chemical vapor deposition (AP-CVD) route (see Methods and Supplementary Fig. 1). Figure 1a illustrates the structural transformation process. After oxygen plasma treatment, the top $WSe_2$ layer can be converted to the preferentially arranged 1D $WO_{3-x}$ nanowires, indicated by the inset transmission electron microscopy (TEM) image, which is highly reproducible (Supplementary Figs. 2 and 3). Firstly, we investigated such 2D $WSe_2$-to-1D $WO_{3-x}$ conversion process on monolayer samples by regulating the plasma treatment durations and found that the 60 s treatment time is enough for the final

transformation. (Supplementary Figs. 4–6). Then, 1D/2D $WO_{3-x}$/$WSe_2$ heterostructures were constructed by exposing the few-layer $WSe_2$ to oxygen plasma for 60 s. Optical images show that the contrast of pristine $WSe_2$ slightly shallows after the top layer was oxidized to $WO_{3-x}$ (Fig. 1b, c), consistent with the phenomena of oxidized monolayer $WSe_2$. Pristine $WSe_2$ exhibits an atomically flat surface according to the atomic force microscope (AFM) topographic image (Fig. 1d). Interestingly, there are 1D $WO_{3-x}$ nanowires stretched from the edge after forming $WO_{3-x}$/$WSe_2$ heterostructures (Fig. 1e), which are parallel or at an angle of 120° to the edge, following the three-fold symmetry of bottom $WSe_2$. The AFM images of heterostructures obtained by different plasma durations (e.g., 15 s, 30 s, 45 s) were also presented, showing similar oriented alignment relationship (Supplementary Fig. 7). According to the statistic length and diameter distributions of $WO_{3-x}$ nanowires as the function of the plasma treatment durations (Supplementary Figs. 8 and 9), the lengths and diameters of $WO_{3-x}$ nanowires gradually decrease with increasing the treatment durations. This can be explained that prolonging plasma durations could produce more Se vacancies, which would induce more freedom degrees of linear arrangement (see Supplementary Part 10). Thus, Se vacancies will form shorter and narrower linear alignment confined at a flake, leading to the final $WO_{3-x}$ nanowires with relatively short lengths and narrow diameters. In addition to treatment duration, the plasma frequency is one key parameter for the formation of $WO_{3-x}$ nanowires (Supplementary Fig. 10). The AFM profile indicates that the thickness of an individual $WO_{3-x}$ nanowire is nearly 2.0 nm (Fig. 1f).

Raman and photoluminescence (PL) spectra were used to analyze the structures and optical properties of samples. The Raman spectrum of pristine $WSe_2$ shows two characteristic peaks of $E_{2g}$ at ~249 cm$^{-1}$ and $A_{1g}$ at ~258 cm$^{-1}$ (Fig. 1g)[30], respectively. The Raman peak of $B_{2g}$ at ~304 cm$^{-1}$ can be attributed to the interlayer interactions[31], indicating the nature of few-layer $WSe_2$. After forming $WO_{3-x}$/$WSe_2$ heterostructures, the $B_{2g}$ mode of $WSe_2$ is still detected besides $E_{2g}$ and $A_{1g}$, indicating the preservation of the underlying $WSe_2$ layers. The pristine $WSe_2$ shows prominent PL peaks at ~782 nm and ~860 nm (Fig. 1h), corresponding to the direct and indirect bandgaps of trilayer $WSe_2$, respectively. For $WO_{3-x}$/$WSe_2$ heterostructures, the PL peaks shift to ~779 nm and ~812 nm, which can be assigned to the bilayer emission[32]. According to the X-ray photoelectron spectroscopy (XPS) (Fig. 1i), there are $W^{6+}$ $4f$ and $W^{5+}$ $4f$ chemical states except for the $W^{4+}$ $4f$ peaks in the $WO_{3-x}$/$WSe_2$ heterostructures compared to that in pristine $WSe_2$, which can be attributed to the formation of $WO_{3-x}$.

The synthesis of $WO_{3-x}$ nanowires from pristine $WSe_2$ was further investigated by microscopy analysis. Figure 2a shows the morphology of one exposed corner of the triangular-shape $WSe_2$, and the corresponding selected-area electron diffraction (SAED) pattern (Fig. 2b) proves its single crystalline nature. The high-angle annular dark-field scanning transmission electron microscopy (HAADF-STEM) image further shows the well-ordered honeycomb structure of pristine $WSe_2$ (Fig. 2c). In comparison, for the oxygen plasma-treated few-layer $WSe_2$, the top $WSe_2$ layer is oxidized to 1D $WO_{3-x}$ nanowires and formed a woven structure (Fig. 2d). These nanowires are closely arranged, consistent with our AFM results. As evidenced by the SAED pattern (Supplementary Fig. 11a), the underlying $WSe_2$ layers still reserve the pristine form during the plasma treatment[33], where two sets of diffraction spots corresponding to the underlying $WSe_2$ (yellow dotted circle) as well as the top $WO_{3-x}$ (green dotted circle) can be also observed. Additionally, the crystalline structure of the underlying $WSe_2$ was verified to keep the pristine form after immersing the $WO_{3-x}$/$WSe_2$ heterostructures into 1 M KOH etchant to remove the top $WO_{3-x}$ nanowires (Supplementary Figs. 13–16). The 1D $WO_{3-x}$ nanowires are preferentially and energetically favorable to arrange along the three-fold symmetric directions of the underlying $WSe_2$ (Fig. 2e). Such preferential distribution of $WO_{3-x}$ nanowires can also be clearly identified based on the TEM and energy dispersive X-ray spectroscopy (EDS)

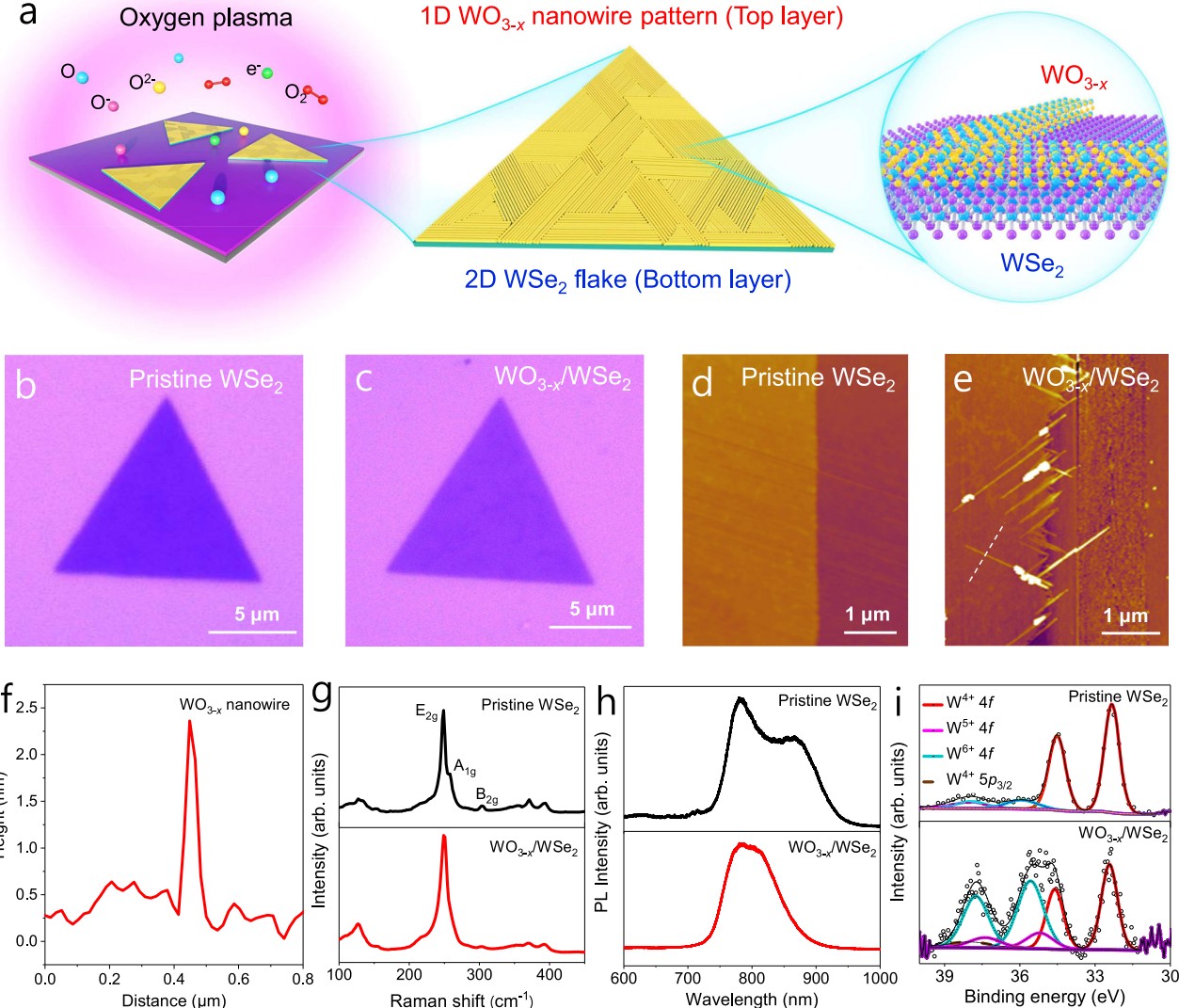

**Fig. 1 | Formation and morphology of 1D/2D WO$_{3-x}$/WSe$_2$ heterostructures.**
**a** Schematic illustration of the synthesis of 1D oriented WO$_{3-x}$ nanowires from 2D WSe$_2$ flake via oxygen plasma treatment. **b**, **c** Optical images of pristine WSe$_2$ and WO$_{3-x}$/WSe$_2$ heterostructures. **d**, **e** Atomic force microscope (AFM) images of pristine WSe$_2$ and WO$_{3-x}$/WSe$_2$ heterostructures. **f** Height profile of an individual WO$_{3-x}$ nanowire corresponding to the white dotted line in (**e**). **g** Raman spectra, **h** photoluminescence spectra, and **i** X-ray photoelectron spectroscopy (XPS) fine scan spectra of pristine WSe$_2$ and WO$_{3-x}$/WSe$_2$ heterostructures.

elemental mapping results observations (Supplementary Figs. 11b, c and 12). Moreover, Fig. 2f shows the interface microstructure of WO$_{3-x}$/WSe$_2$ heterostructures at atomic scale. Considering that the lattice mismatch at the interfaces of WO$_{3-x}$ nanowires can locally introduce strain in order to maintain the thermodynamic stability of the three-fold symmetric alignment (Supplementary Fig. 17a), here we use geometric phase analysis (GPA) to map the strain field at the junction interface (Supplementary Fig. 17b–f). Such lattice distortion can significantly alter the local electronic structures, which might influence the charge transfer process in SERS[34]. All these results demonstrate that the oxygen plasma treatment is self-limited in nature, inducing the formation of 1D/2D WO$_{3-x}$/WSe$_2$ heterostructures.

In order to further investigate the structural transformation process of 2D WSe$_2$-to-1D WO$_{3-x}$ nanowires, few-layer 2D WSe$_2$ was treated by oxygen plasma only for seconds for observations (Supplementary Fig. 18). At the start of the treatment, top WSe$_2$ layer was etched into notches with triangular-like and hexagonal-like shapes, forming separated amorphous clusters (Supplementary Fig. 19). As the reaction continued, these separated clusters are reconstructed into crystalline WO$_{3-x}$ nanowires through a crystallization process (Fig. 2g). It is worth mentioning that for these few-layer WSe$_2$ samples, the layer numbers of WSe$_2$ around WO$_{3-x}$ nanowires are usually one layer thinner than that of parent materials (Supplementary Fig. 20), which proves the etching effect of oxygen plasma. Also, there are numbers of amorphous clusters attached at the edge or end of the as-formed WO$_{3-x}$ nanowires, presenting an intermediate state of the transformation due to the short treatment time (Fig. 2g). Density functional theory (DFT) calculations were conducted to illustrate this structural transformation process (See details in Supplementary Figs. 21–26). Thus, the structural transformation process of 2D WSe$_2$-to-1D WO$_{3-x}$ nanowires are well elucidated both in experiment and theory.

## Molecular sensing performance based on SERS effect

Figure 3a illustrates the Raman scattering process of MB molecules on WO$_{3-x}$/WSe$_2$ heterostructures excited by 633 nm laser. As control experiment, the Raman signals collected from the bare SiO$_2$/Si and pristine WSe$_2$ are shown in Supplementary Fig. 27. Raman signals of MB molecules on pristine WSe$_2$ (marked by "♥") show stronger intensity than those on bare SiO$_2$/Si substrate, indicating the obvious SERS effect of WSe$_2$. Figure 3b shows the Raman spectra of MB

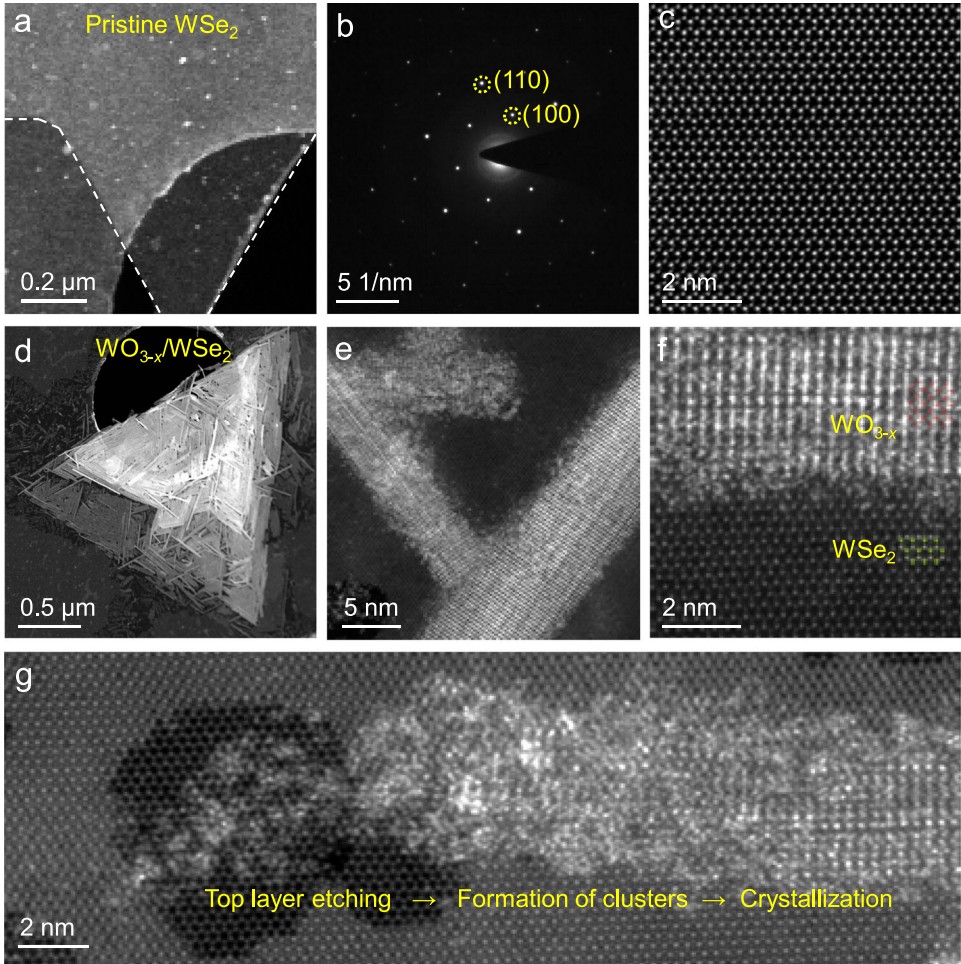

**Fig. 2 | Transmission electron microscope (TEM) characterizations of pristine WSe₂ and 1D/2D WO₃₋ₓ/WSe₂ heterostructures. a** TEM image of pristine WSe₂ supported on a holey carbon-coated TEM grid. **b, c** Selected-area electron diffraction (SAED) pattern (**b**) and high-angle annular dark-field scanning TEM (HAADF-STEM) image (**c**) of the pristine WSe₂. **d–f** HAADF-STEM images of WO₃₋ₓ/ WSe₂ heterostructures at different magnifications. Gray, green, and red spheres in (**f**) represent W, Se, and O atoms, respectively. **g** HAADF-STEM image of a bilayer WSe₂ treated by oxygen plasma for only 5 s in order to capture the intermediate states. Partial WSe₂ was etched and formed amorphous clusters, which were then crystallized into WO₃₋ₓ nanowires under the plasma atmosphere.

molecules on monolayer WSe₂ treated by different oxygen plasma durations and the Raman intensity at ~1620 cm⁻¹ are shown in Supplementary Fig. 28. The strongest Raman signals of MB molecules can be obtained when WSe₂ was completely converted to 1D WO₃₋ₓ nanowires after 60 s plasma treatment. Raman spectra taken from ~20 random points were collected to evaluate the uniformity of MB molecules on WO₃₋ₓ (Fig. 3c), which are nearly identical with a small relative standard deviation (RSD, 16%, Supplementary Fig. 29), indicating the uniform SERS effect of WO₃₋ₓ and the homogeneous adsorption of MB molecules on WO₃₋ₓ. While for the Au nanoparticles (Au NPs) substrates synthesized using a standard sodium citrate reduction method[35], the Raman intensity of MB molecules at various sites shows huge differences (RSD 72%, Supplementary Fig. 30).

The LOD of SERS substrates is a vital indicator to evaluate the sensitivity. Therefore, a series of MB solutions with concentrations from $5 \times 10^{-6}$ to $5 \times 10^{-18}$ M were prepared. The lowest detection concentration of WSe₂ for MB molecules is only $5 \times 10^{-8}$ M (Supplementary Figs. 31–32), while the LOD of MB molecules on WO₃₋ₓ can reach a low level of $5 \times 10^{-12}$ M (Fig. 3d). Even increasing the integration time from 5 s to 40 s, the Raman fingerprints of MB molecules with the concentration of $5 \times 10^{-13}$ M on 1D WO₃₋ₓ cannot be obtained (Supplementary Fig. 33). WO₃₋ₓ nanoflakes (WO₃₋ₓ-NF) were synthesized based on our previous study to verify the dimensional effect (Supplementary Fig. 34)[36]. The Raman intensity of MB on 1D WO₃₋ₓ is nearly 10 times

stronger than that on WO₃₋ₓ-NF (Supplementary Fig. 35), demonstrating the effective enhancement of the dimensionality on the SERS effect of WO₃₋ₓ nanowire patterns. The adsorption behaviors of MB molecules on WSe₂ and WO₃₋ₓ were comparably investigated by DFT calculations to elucidate their SERS effects. As shown in Fig. 4a, b, the adsorption energy for MB molecule on the WO₃₋ₓ substrate is −4.82 eV, which is obviously stronger than that on the WSe₂ substrate (−1.63 eV). Besides, an efficient charge transfer of 0.39 e between the MB molecule and the WO₃₋ₓ substrate can induce a stronger interfacial dipole of MB/WO₃₋ₓ than that of MB/WSe₂ (0.11 e). Therefore, the sensitive SERS effect of WO₃₋ₓ can be attributed to the following three aspects: (1) the interfacial lattice distortion of the WO₃₋ₓ nanowire patterns will facilitate the charge transfer between substrate and molecules[37,38], which can be demonstrated by the DFT calculations with the increased charge transfer to 0.40 e after applying biaxial strain to the WO₃₋ₓ nanowire (Supplementary Fig. 36a). (2) The 1D nanowire structure of WO₃₋ₓ facilitates the efficient electron transport along the axis directions[39]. (3) The existence of oxygen vacancies in WO₃₋ₓ can also enhance the interactions with probe molecules via vibronic coupling to further enhance the charge transfer efficiency[40]. Therefore, the excellent molecular sensing performance of WO₃₋ₓ nanowire patterns is attributed to the synergistic effect of the above-mentioned factors.

More importantly, the Raman signals of MB molecules on 1D/2D WO₃₋ₓ/WSe₂ heterostructures are still detectable when the MB solution

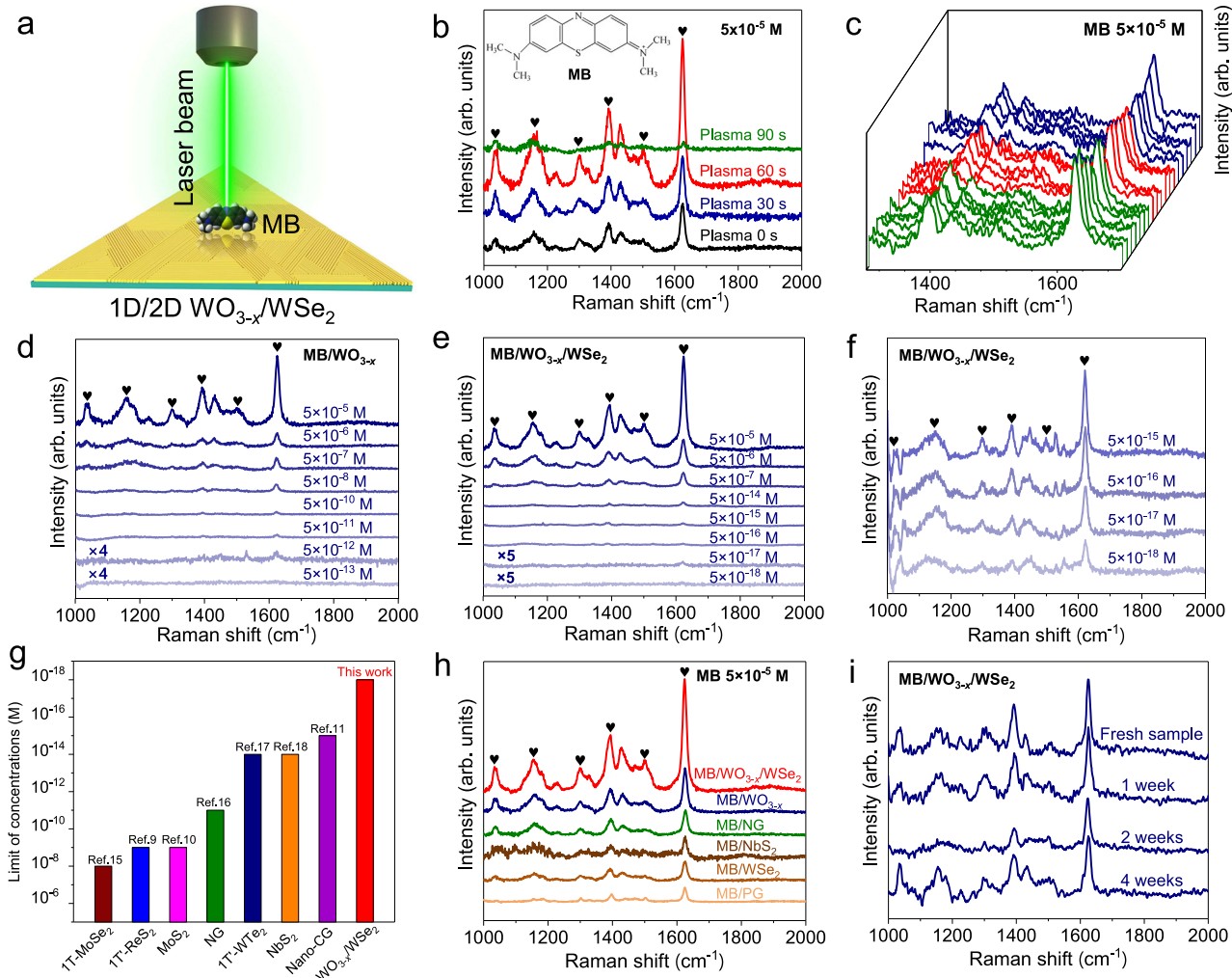

**Fig. 3 | Surface-enhanced Raman scattering (SERS) performance of 1D/2D WO₃₋ₓ/WSe₂ heterostructures. a** Schematic illustration of Methylene Blue (MB) molecules on WO$_{3-x}$/WSe$_2$ heterostructures excited by Raman laser beam. **b** Raman spectra of MB on WSe$_2$ with different oxygen plasma treatment durations. Inset shows the chemical structure of MB. "♥" symbols represent the characteristic Raman peaks of MB molecules. **c** Raman spectra of MB collected from ~20 spots on WO$_{3-x}$ nanowires. **d–f** Raman spectra of MB on WO$_{3-x}$ nanowires (**d**) and WO$_{3-x}$/WSe$_2$ heterostructures (**e**, **f**) with different concentrations. **g** Limit of detections

(LODs) comparison of the state-of-the-art 2D materials: 1T-MoSe$_2$[15], 1T′-ReS$_2$[9], 1H-MoS$_2$[10], nitrogen-doped graphene (NG)[16], 1T′-WTe$_2$[17], NbS$_2$[18], and nanocorrugated graphene (Nano-CG)[11]. **h** SERS effects of MB on various 2D materials. **i** Raman spectra of MB on WO$_{3-x}$/WSe$_2$ heterostructures after exposure in ambient conditions for different durations. Except that the integration time of Raman spectra in (**f**) is 40 s, the integration times for other Raman spectra are all 5 s. WO$_{3-x}$ refers to WO$_{3-x}$ nanowire patterns unless stated.

is down to a low concentration of $5 \times 10^{-18}$ M as shown in Fig. 3e, which is 3–10 orders of magnitude lower than those of the state-of-the-art 2D materials and comparable or even superior to those of noble-metal-based substrates (Fig. 3g, Supplementary Fig. 37 and Supplementary Table 1). For the ultralow concentrations of MB from $5 \times 10^{-15}$ to $5 \times 10^{-18}$ M, the integration time for each Raman spectrum was prolonged to 40 s to clearly identify the Raman fingerprints of MB in Fig. 3f. The EF of WO$_{3-x}$/WSe$_2$ was calculated to be $5.0 \times 10^{11}$ (bulk MB as reference, Supplementary Fig. 38). Even in the mixed solution of MB ($5 \times 10^{-18}$ M) and R6G ($5 \times 10^{-10}$ M) with the 1:1 volume ratio, the Raman fingerprints of MB can also be clearly detected (Supplementary Fig. 39), indicating the ultrasensitive SERS effect of WO$_{3-x}$/WSe$_2$ heterostructures even under the interference of other molecules. While, the LOD of MB on the 2D/2D WO$_{3-x}$-NF/WSe$_2$ is $5 \times 10^{-14}$ M (Supplementary Fig. 40), which is inferior to that of 1D/2D WO$_{3-x}$/WSe$_2$ benefiting from its unique structure of the preferentially arranged 1D nanowires.

The SERS effect of WO$_{3-x}$/WSe$_2$ heterostructures can also be extended to detect other dye molecules (e.g., rhodamine 6 G (R6G),

crystal violet (CV)). The Raman fingerprints of R6G and CV molecules on WO$_{3-x}$/WSe$_2$ are clearly distinguished (Supplementary Fig. 41a, b), as marked by the symbols of "♣" and "♦", respectively. And, the LODs of WO$_{3-x}$/WSe$_2$ for the detection of R6G and CV are superior to most of the reported non-noble-metal substrates (Supplementary Table 1). Even compared with Au NPs substrates, WO$_{3-x}$/WSe$_2$ heterostructures still exhibit much higher sensitivity (Supplementary Fig. 41c, d). The different LODs for the detection of various dye molecules on WO$_{3-x}$/WSe$_2$ heterostructures might be due to the different binding energy and charge transfer process between the substrates and the probe molecules, and different photo-induced charge transfer (PICT) process from the band edges of substrates to the affinity levels of probe molecules. Such demonstrated universal SERS effect of WO$_{3-x}$/WSe$_2$ will endow it promising potentials in many fields, including food safety, chemical analysis, environmental monitoring, etc. Meanwhile, by choosing different laser lines, Raman fingerprints of individual MB and R6G in the mixed solution can be clearly detected on WO$_{3-x}$/WSe$_2$ (Supplementary Fig. 42), respectively, demonstrating the excellent selectivity of WO$_{3-x}$/WSe$_2$ substrate.

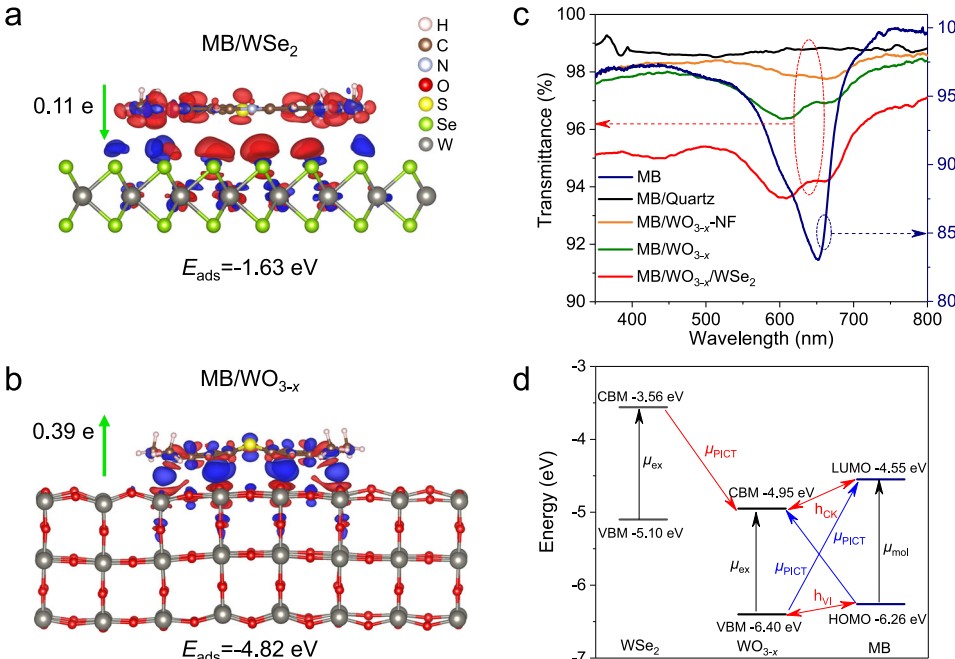

**Fig. 4 | Chemical enhancement mechanism of molecular sensing. a, b** Charge density differences of MB molecules on the $WSe_2$ (**a**) and $WO_{3-x}$ (**b**). Blue (red) corresponds to the charge accumulation (depletion). $E_{ads}$ represents the adsorption energy for MB adsorbed on sample substrates. The isosurface values of MB adsorbed on $WSe_2$ and $WO_{3-x}$ are 0.00005 and 0.0003 $e/A^3$, respectively. **c** Ultraviolet (UV)-visible transmission spectrophotometry of MB/quartz, MB/$WO_{3-x}$ nanoflakes (MB/$WO_{3-x}$-NF), MB/$WO_{3-x}$ and MB/$WO_{3-x}$/$WSe_2$. **d** Energy level alignment and photo-induced charge transfer (PICT) transition in the MB/$WO_{3-x}$/$WSe_2$. VBM: valence band maximum. CBM: conduction band minimum. HOMO: highest occupied molecular orbital. LUMO: lowest unoccupied molecular orbital. $\mu_{ex}$: exciton transition. $\mu_{mol}$: Molecular transition. $\mu_{PICT}$: PICT transition. $h_{CK}$, $h_{VI}$: Herzberg-Teller coupling constant.

Ultraviolet (UV)-visible transmission measurements were carried out to investigate the interactions between the MB and different samples (Fig. 4c). Due to the weak adsorption, the absorption peak of MB in the visible range is hardly detected on quartz. In stark contrast, the absorption peak of MB molecules can be obviously observed upon adsorbed on $WO_{3-x}$ and $WO_{3-x}$/$WSe_2$. Meanwhile, the relative intensity and location of MB peaks on $WO_{3-x}$ and $WO_{3-x}$/$WSe_2$ are remarkably changed compared to those of pristine MB, demonstrating the strong adsorption of MB onto $WO_{3-x}$ and $WO_{3-x}$/$WSe_2$, which can further facilitate the efficient electron transition probability rate between them. Moreover, the energy level alignment between MB and $WO_{3-x}$/$WSe_2$ heterostructures was calculated by DFT (Fig. 4d). For the MB/$WO_{3-x}$ system, the highest occupied molecular orbital (HOMO) and lowest unoccupied molecular orbital (LUMO) levels of MB are at −6.26 and −4.55 eV, respectively. The valence band maximum (VBM) and conduction band minimum (CBM) of $WO_{3-x}$ are located at −6.40 and −4.95 eV, respectively. The molecular transition ($\mu_{mol}$ = 1.71 eV) and exciton transition ($\mu_{ex}$ = 1.45 eV) in the MB/$WO_{3-x}$ system can be excited by 633 nm laser to promote the Raman scattering. And, the PICT transition (1.85 eV) from $WO_{3-x}$ VBM to MB LUMO are beneficial to the enhancement of the SERS effect due to the charge transfer resonance, which can borrow intensity from the molecular transition and exciton transition through the Herzberg-Teller coupling constant ($h_{VI}$ and $h_{CK}$) to make the probe molecules much more polarized and further increase the Raman scattering cross-section[41]. The molecular transition resonance can be verified by exciting the Raman signals of MB on sample with different laser lines (Supplementary Fig. 43). When excited by 633 nm laser, the MB molecules demonstrate the strongest Raman intensity compared with those by other laser lines (e.g., 532 nm, 785 nm), since 633 nm laser possesses the nearest energy with the energy level difference (1.71 eV) between LUMO and HOMO of MB. Meanwhile, a charge transfer from the CBM of $WSe_2$ to the CBM of $WO_{3-x}$ can occur (Fig. 4d). In order to elucidate the effect of electron doping on the SERS effect of $WO_{3-x}$/$WSe_2$ heterostructures, the surface

W atoms in $WO_{3-x}$ are doped by electrons. The calculated adsorption energy decreases to −5.05 eV, and the charge transfer increases to 0.40 e for the MB on $WO_{3-x}$/$WSe_2$ heterostructures. Under the synergetic effect of electron doping and biaxial strain, the charge transfer can further increase to 0.42 e (Supplementary Fig. 36b), which evidences that the charge transfer transition from $WSe_2$ to $WO_{3-x}$ and the strain between $WO_{3-x}$ nanowires can synergistically enhance the SERS performance of $WO_{3-x}$ after constructing heterostructures with $WSe_2$.

To accurately quantify the charge transfer process, systematic ultrafast transient absorption/transmission experiments were carried out. In the steady-state spectra (Fig. 5a), the MB absorption spectrum on $WO_{3-x}$/$WSe_2$ emerges as broadened intensity increase/peak from ~600 to ~700 nm, consistent with the transmission results shown in Fig. 4c, though the light source here is femtosecond (fs) white light continuum (see "Methods"). Meanwhile, the A exciton (750 nm) of $WSe_2$ is effectively suppressed, while its B exciton remains unchanged. In Fig. 5b and Supplementary Fig. 47, the transient transmission spectrum of MB/$WO_{3-x}$/$WSe_2$ shows a pump-induced bleaching peak (positive peak in ΔT/T curves) at ~690 nm, at the expense of the $WSe_2$ A exciton. These phenomena have not been found in MB/$WSe_2$ or $WO_{3-x}$/$WSe_2$, suggesting that $WO_{3-x}$ plays an intermediary role in the ultrafast charge transfer processes in the form of MB/$WO_{3-x}$/$WSe_2$. This observation strongly supports our proposal in Fig. 4d. More details in Fig. 5c indicate that the bleaching peak of MB (representative curves at 690 nm) is about 1.0 ps delay after the $WSe_2$ A exciton peak, for the case of resonant pumping by 750 nm. We notice that by pumping with 532 and 633 nm, the bleaching curves of MB first present a peak at ~0.5 ps (nearly identical to that of the $WSe_2$ A exciton peak due to the thermalization process, i.e., high-energy hot carriers relax to the bottom of conduction bands), followed by a slope change/second peak after 1.0 ps. This can be explained that by over-gap pump, both direct excitation of the bandgap and charge transfer contribute to the pump-induced bleaching feature of MB in different timescales. These dynamics are much more clearly resolved in secondary derivative

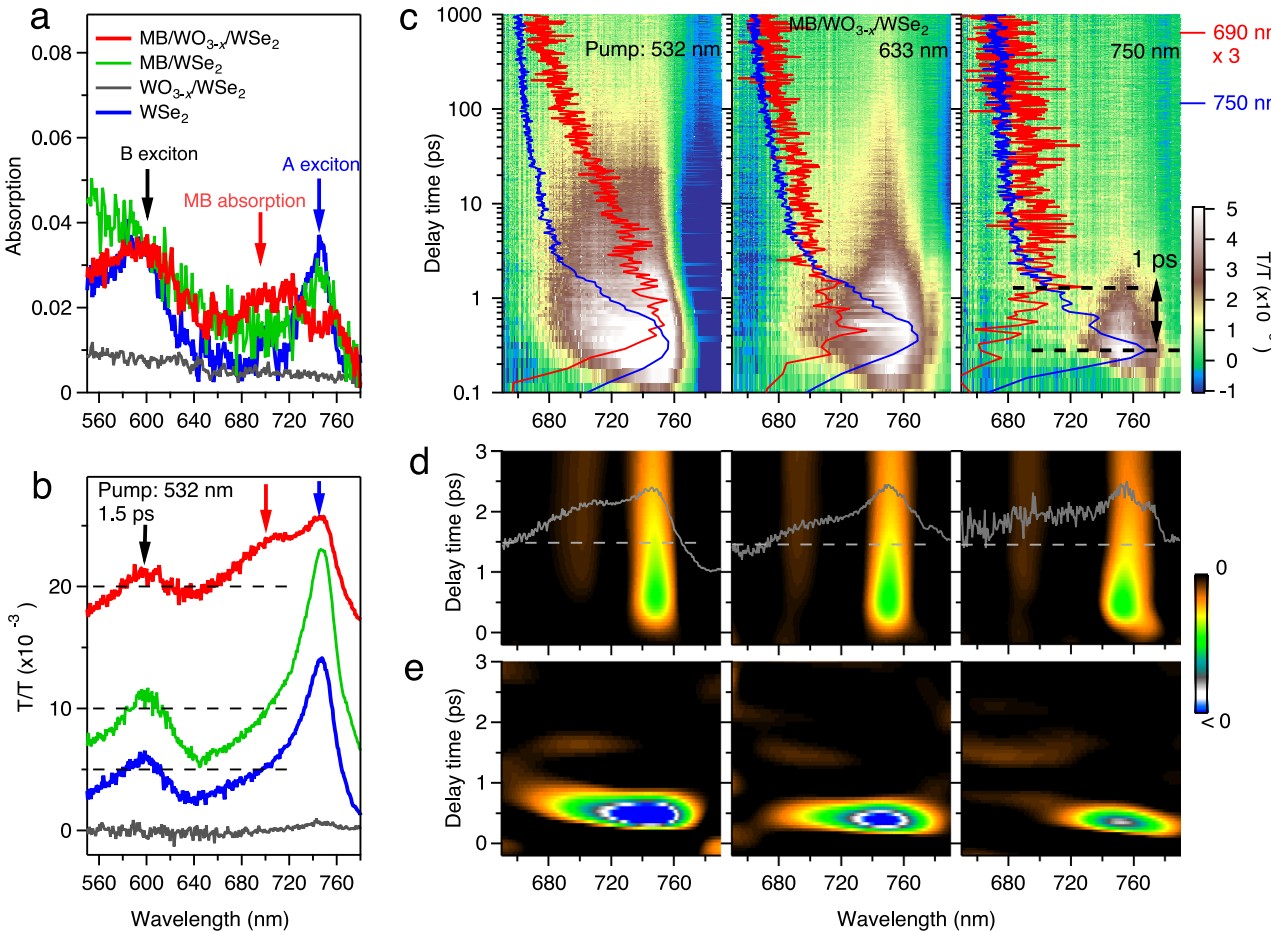

**Fig. 5 | Ultrafast transient absorption/transmission spectra of MB/WO$_{3-x}$/WSe$_2$.**
**a** Steady-state optical absorption spectra of MB/WO$_{3-x}$/WSe$_2$, MB/WSe$_2$, WO$_{3-x}$/WSe$_2$, and WSe$_2$. **b** Transient transmission curves of MB/WO$_{3-x}$/WSe$_2$ at a fixed delay time $t = 1.5$ ps with a pump wavelength of 532 nm. The dashed zero lines are shifted together with the corresponding data curves. **c** Transient transmission images of MB/WO$_{3-x}$/WSe$_2$ with different pump wavelengths of 532 nm, 633 nm, and 750 nm. The time-dependent curves at 690 nm and 750 nm are in red and blue, respectively. The black arrow depicts the charge transfer timescale of about 1.0 ps. **d, e** Secondary derivative images with respect to the wavelength (**d**) and the delay time (**e**), corresponding to the original data in (**c**). The gray solid curves in (**d**) are wavelength-dependent data at 1.5 ps, plotted with the dashed zero lines. Full comparisons are shown in Supplementary Fig. 47.

images shown in Fig. 5d, e (Supplementary Fig. 48). In addition, the transient transmission spectroscopy of the MB adsorbed on 2D/2D WO$_{3-x}$-NF/WSe$_2$ was carried out for comparison (Supplementary Fig. 49). The MB bleaching signal on WO$_{3-x}$-NF/WSe$_2$ is nearly an order of magnitude weaker than that on 1D/2D WO$_{3-x}$/WSe$_2$, suggesting that the 1D/2D WO$_{3-x}$/WSe$_2$ heterostructures exhibits an enhanced charge transfer process by almost a factor of ten. Overall, the ultrafast spectroscopy results indicate a charge transfer process between WO$_{3-x}$/WSe$_2$ heterostructures and MB at the timescale of ~1.0 ps.

A series of control experiments were implemented to compare the SERS effects between WO$_{3-x}$/WSe$_2$ heterostructures and other 2D materials. These 2D materials were synthesized by AP-CVD routes based on our previous works (Supplementary Fig. 44)[16,42]. The strongest Raman intensity of MB molecules can be excited on WO$_{3-x}$/WSe$_2$ heterostructures (Fig. 3h, Supplementary Fig. 45). The weak Raman signals of MB on NbS$_2$ might be ascribed to the adsorption of impurities and the oxidation of metallic NbS$_2$ in air. Nitrogen-doped graphene (NG) presents a higher SERS effect than that of pristine graphene (PG), suggesting that defect engineering significantly influences the SERS effect. Furthermore, the stability of WO$_{3-x}$/WSe$_2$ heterostructures was evaluated (Fig. 3i, Supplementary Fig. 46). Even after exposure in air for a much longer time (4 weeks), there is only a slight degradation (~10% loss) of Raman intensity of MB compared to that on fresh sample. The excellent stability of heterostructures is superior to the metal/semimetal TMDCs[17,18], which is important for the practical

applications. In general, the superior sensitivity, well-resolved Raman fingerprint peaks, excellent selective detection capability, and good stability against the air demonstrate 1D/2D WO$_{3-x}$/WSe$_2$ heterostructures exhibit promising potentials for the practical applications in the future.

## Discussion

In summary, we demonstrate a concept of 1D nanowire patterns evolved from 2D flakes and their 1D/2D heterostructures by a facile and effective oxygen plasma approach. Take 2D WSe$_2$ as an example, as-obtained 1D WO$_{3-x}$ nanowires are preferentially and energetically arranged along the three-fold symmetric directions of the parental WSe$_2$. An ultrasensitive molecular sensing performance with a low LOD of $5 \times 10^{-18}$ M and a high EF of $5.0 \times 10^{11}$ for MB molecules is achieved on the WO$_{3-x}$/WSe$_2$ heterostructures, even in the mixed solution with other molecules. This ultralow LOD of WO$_{3-x}$/WSe$_2$ heterostructures is 3–10 orders of magnitude lower than those of the state-of-the-art noble-metal-free 2D materials and comparable or even superior to those of noble-metal-based substrates. The charge transfer timescale between the heterostructures and the probe molecules can be clearly unveiled at ~1.0 ps by the ultrafast pump-probe transient spectroscopy. The WO$_{3-x}$/WSe$_2$ heterostructures also show excellent selectivity, stability, and universal SERS effects for probing other molecules. Our study provides alternative insights into the synthesis of oriented 1D nanowire patterns from 2D materials and prospects for the

development of unique mixed-dimensional structures by a facile, effective, and scalable route, which can be applied in diverse fields (e.g., clinical diagnosis, neuromorphic devices, energy storage/conversion). Furthermore, our work also demonstrates the importance of ultrafast transient spectroscopy in SERS fields to unveil the underlying mechanism.

## Methods

### Sample synthesis
Monolayer and few-layer $WSe_2$ flakes were grown in the AP-CVD system. $WO_3$ powder and Se powder were used as W and Se precursors, respectively. NaCl was added to assist the growth of $WSe_2$. The weight ratios of $WO_3$ and NaCl were set as 8:1 and 8:3 for the growth of monolayer $WSe_2$ and few-layer $WSe_2$, respectively. The solution of W precursor was prepared by dissolving $WO_3$ powder and NaCl powder into 10 mL ammonia solution under stirring at 80 °C for 1.5 h. Then, a drop of the solution (~5 μL) was spin-coated onto a piece of $SiO_2$/Si substrate at 3000 rpm for 60 s. For the growth of $WSe_2$, the $SiO_2$/Si substrate spin-coated with the precursor film was placed in a quartz boat and loaded into the center of the CVD reactor. Se powder (~800 mg) was placed at another quartz boat and placed at upstream, ~5 cm away from the inlet of the reactor (See Supplementary Fig. 1). Before heating, the quartz tube reactor was purged with ~1500 sccm Ar for ~5 min. The furnace was heated to 850 °C with Ar at a flow rate of 80 sccm, and maintained for 6 min to grow $WSe_2$ with Ar/$H_2$ (80/6 sccm) flow. Finally, the sample was quickly cooled to the room temperature by pulling the quartz tube out of the furnace.

### Synthesis of 1D $WO_{3-x}$ nanowires and 1D/2D $WO_{3-x}$/$WSe_2$ heterostructures
The lattice reconstruction process of $WSe_2$ was carried out in an oxygen plasma system (Schwarze). The oxygen plasma was generated with the frequency of 40 kHz. For the synthesis of 1D $WO_{3-x}$ nanowires, monolayer $WSe_2$ was placed in the center of the chamber and treated for 2–90 s by oxygen plasma. For the synthesis of heterostructures, few-layer $WSe_2$ was treated by oxygen plasma with the same duration as that of monolayer $WSe_2$.

### Sample characterization
Optical images were captured by an Olympus BX 53 M microscope. AFM images were recorded using an Oxford MFP-3D Infinity system in a tapping mode. XPS (Thermo Fisher ESCALAB 250Xi) was used to analyze the chemical states. The binding energies were calibrated with C 1 s at 284.8 eV. SEM images were collected on a Zeiss Merlin system. TEM analysis was performed on FEI Tecnai F30 system and JEOL JEM-2100 system, operating at 300 kV and 200 kV accelerating voltages, respectively. HAADF-STEM images were taken on a FEI Spectra 300, operating at 80 kV accelerating voltage. UV-visible transmission measurements were recorded on U-3900H Spectrophotometer at a scan speed of 120 nm min$^{-1}$.

### SERS measurements
MB, R6G, and CV molecules were dissolved in ethanol to form solutions with the initial concentration of $5 \times 10^{-5}$ M. Then, the dye solutions with different concentrations ranging from $5 \times 10^{-6}$ to $5 \times 10^{-18}$ M can be prepared by sequential dilution process. Samples grown on $SiO_2$/Si substrates ($0.5 \times 0.5$ cm$^2$) were immersed into the solutions with different concentrations for ~30 min, respectively, followed by naturally drying. Then, the samples were rinsed with ethanol to remove the free molecules and dried with $N_2$ gas. All the Raman spectra were collected in a HORIBA LabRAM HR system with the ×50 objective. The Raman excitation wavelength for probing MB is 633 nm but for probing R6G and CV is 532 nm. The laser power is below 1 mW to avoid the possible heating effect on samples. The integration time for MB, R6G, and CV with the concentrations from $5 \times 10^{-5}$ to $5 \times 10^{-14}$ M is 5 s, while

for MB with the concentrations from $5 \times 10^{-15}$ to $5 \times 10^{-18}$ M is 40 s. The Raman spectra were collected with the same experimental parameters for the better comparison among different 2D materials and noble-metal substrates.

### Ultrafast transient absorption/transmission measurements
The ultrafast pump-probe measurements were conducted in a transient absorption/transmission spectrometer (Helios fire from Ultrafast system) at room temperature. The samples were transferred onto the transparent sapphire substrates. The infrared pulses (800 nm, 35 fs) were provided by a Ti:Sapphire amplifier (Coherent Inc.) working at 1 kHz repetition rate, and splitted into two beams. One went to an optical parameter amplifier (Opera Solo System from Coherent Inc.) to produce tunable pumps (532 nm, 633 nm, and 750 nm), while the other one was delayed by a linear stage and focused into a sapphire crystal to generate white-light continuum as probe. The pump and probe beams were focused on sample surface with spot sizes of about 200 μm and 7 μm, respectively, by applying the parabolic reflectors. All measurements were taken with a constant pump fluence of 38 μJ cm$^{-2}$. A back-thinned CCD linear detector synchronized with an optical chopper was applied to measure the transient transmission difference, through the detection of probe signals with and without pump.

### DFT calculations
All calculations based on density functional theory (DFT) were performed using the Vienna ab initio simulation package (VASP)[43]. The projector augmented wave (PAW) potentials[44] and generalized gradient approximation (GGA) of the Perdew–Burke–Ernzerhof (PBE) functional[45] were used for the electron-ion interaction and exchange-correlation energy, respectively. An $8 \times 8 \times 1$ ($7 \times 4 \times 1$) supercell for $WSe_2$ ($WO_{3-x}$) nanosheet was built to investigate the adsorption behaviors of MB molecules. The cutoff energy of the plane wave basis was set to 400 eV. The convergence criteria for the total energy and force were set to $10^{-5}$ eV and 0.01 eV Å$^{-1}$, respectively. The vacuum layer of at least 15 Å was chosen to eliminate the interactions between the periodic images. The DFT-D3 method[46] was used to consider the van der Waals interaction between MB molecules and sample substrates. The adsorption energy for MB adsorbed on sample substrates was calculated using $E_{ads} = E_{MB/sample} - E_{sample} - E_{MB}$, where $E_{MB/sample}$, $E_{sample}$, and $E_{MB}$ are the energies of MB adsorbed on sample substrates, sample substrates, and MB molecules, respectively. Because the states near the Fermi level of $WO_{3-x}$ were mainly originated from p orbitals of W, the ionic potential method was used to dope electrons on W atoms from W 3d core level to the lowest unoccupied band. This method ensured that the doped electrons were localized around the W atoms and maintained the neutrality of the sample.

## Data availability
The Source data underlying the figures of this study are available at https://doi.org/10.6084/m9.figshare.22565134. All raw data generated during the current study are available from the corresponding authors upon request.

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

## Acknowledgements

This work was supported by the National Natural Science Foundation of China with Grant No. 51972191 (R.L.), the National Key Research and Development Program of China with Grant Nos. 2021YFA1200800 (R.L.), and 2022YFA1204700 (Q.X.), the National Natural Science Foundation of China with Grant Nos. 11874036 (J.L.), 51920105002 (B.L.), 92056204 (H.L.), and 51991343 (B.L.), the Shenzhen Basic Research Project with Grant Nos. JCYJ20200109142816479 (J.L.), WDZC20200819115243002

(J.L.), and JCYJ20200109144616617 (B.L.) and Local Innovative and Research Teams Project of Guangdong Pearl River Talents Program with Grant No. 2017BT01N111 (J.L.). This work was supported by the Analysis Platform of Center for Nano and Micro Mechanics, Tsinghua University. Computational resources were provided by the Advanced Computing Center of Yunnan University. The authors also acknowledge the use of the TEM facilities at the Institute of Materials Research, Tsinghua Shenzhen International Graduate School (Tsinghua SIGS).

## Author contributions

R.T.L. and Q.H.X. conceived and designed the experiments. Q.L. carried out the main experiments. B.L.L. and J.Y.T. performed the TEM characterization. J.L. and Z.J.W. designed and performed the theoretical calculations. H.Y.L. and P.G. carried out the transient spectroscopy experiment. L.X.Y. synthesized Au nanoparticles. Y.P.W. and L.G. stimulated the strain fields. Q.L., J.Y.T., Z.J.W., H.Y.L., and R.T.L. wrote the manuscript. We thank Prof. F.Y.K. and Prof. H.-M.C. for the fruitful discussion and important advice regarding this work. All authors contributed to data analysis, scientific discussion and commented the manuscript.

## Competing interests

The authors declare no competing interests.
