## [Peer Review File · Nature Communications]

Ultrafast charge transfer in mixed-dimensional WO₃-x nanowire/WSe₂ heterostructures for attomolar-level molecular sensingREVIEWER COMMENTS

Reviewer #1 (Remarks to the Author):

Developing efficient noble-metal-free SERS substrates for ultrasensitive molecular sensing is very meaningful but also very challenging. The authors report the use of a 1D/2D WO_{3-x}/WSe₂ heterostructure as highly-active SERS substrate, which achieves very high sensitivity for MB detection with a record low detection concentration of 5×10⁻¹⁸ M. The experimental results are interesting, and well presented with clear and useful figures. However, some revisions are necessary before the publication of the paper as highlighted below.

(1) It seems that very high sensitivity of 10⁻¹⁸ M is only achieved with MB detection. How about other analytes? Why MB is so special? Considering the need for practical applications, it is better to detect a wide variety of analytes using 1D/2D WO_{3-x}/WSe₂ heterostructure, especially for more analytes in practical uses.

(2) How about the selectivity? Can the SERS substrate show good selectivity in the mixed analyte systems with similar molecular structures?

(3) The authors have emphasized the ultrafast charge transfer of their SERS substrate for sensing. Can the authors compare the charge transfer rates of their SERS substrate and other noble-metal-free and noble-metal SERS substrates using ultrafast transient spectroscopy?

(4) The experimental details for SERS detection should be more detailed. The enhancement factor can be calculated. Commercial metal SERS substrates can be used for comparison.

Reviewer #2 (Remarks to the Author):

Qian Lv et al have constructed a 1D nanowire pattern/2D flake heterostructures in a simple yet effective way, systematically studied the charge transfer effects and the correlation to the SERS activities. The heterostructures were well characterized and physical properties extensively investigated experimentally and theoretically. An extremely high SERS enhancement were achieved in the MB/1D WO_{3-x}/2D WSe₂ heterostructures. They have also uncovered the timescale of around 1.0 picosecond for the charge transfer process with ultrafast transient spectroscopy. It is a quite comprehensive study on charge transfer and SERS effects in this heterostructure. However, it lacks enough novelty nor is of general interest as required by Nature Communications. Some comments and suggestions:

1. The manuscript is entitled "Ultrafast charge transfer of 1D nanowire pattern/2D flake heterostructures for attomolar-level molecular sensing". However, there is no decisive correlation between the ultrafast charge transfer and attomolar-level molecular sensing. Is ~ 1ps transfer time a necessary condition for such ultrahigh SERS sensitivity? What is the physics behind? Do other heterostructures with longer charge transfer time have to be with lower SERS enhancement factor?

2. The authors attribute the strong SERS enhancement of the system to the combined influence of interfacial lattice distortion/strain, efficient electron transport along the axis directions and the existence of oxygen vacancies mediated vibronic coupling, which have already been reported in the literature. No new insight into the enhancement mechanism.

3. The SERS measurements showed that charge transfer between MB and WO_{3-x}/WSe₂ is laser line dependent (Figure S39). Fig 5C indicates that under the resonance effect the charge transfer time scale is about 0.5 ps, whereas the off-resonance charge transfer time scale 1.0 ps, which appears to be caused by a number of effects outside the scope of this manuscript.

4. The authors emphasize the strain effects on charge transfer and thus SERS enhancement in WO_{3-x}/WSe₂. It'd be good that the authors can give the value of the measured strain that might be correlated to the interfacial distance between adjacent layers and thus charge redistribution as revealed by DFT calculation.

5. The manuscript showed nice AFM images of WO_{3-x}/WSe₂ heterostructures. It'd be good if the authors can also present the AFM line scan profiles across the WO_{3-x} nanowires on WSe₂, which can clearly indicate the height of nanowires and 1D/2D stacking morphology.

Reviewer #3 (Remarks to the Author):

This manuscript describes the fabrication of WO_x nanowires on top of few-layer flakes of WSe₂ as a new medium for SERS spectroscopy. The fabrication method is reasonably straightforward and has been characterized with several imaging and spectroscopic methods. The authors then demonstrate SERS activity for the dye molecule methylene blue (MB). They claim that the WO_x nanowires enable an extremely low limit of detection. Finally, the authors use ultrafast spectroscopy in an attempt to probe the mechanism of the SERS enhancement.

Overall, the paper is well written and describes an interesting and promising approach for developing reproducible SERS substrates. In fact, the apparent reproducibility in making the wires (e.g., Fig S2) and especially the apparent reproducibility of the SERS intensity for different samples (Fig 3c) is intriguing. The manuscript has a few inconsistencies and other deficiencies that limit the impact of the work, but these are addressable and a revised version of the manuscript may be suitable for publication so that other researchers can be made aware of the work.

At a minimum, the authors need to address the following concerns (listed in no particular order).

- 1) The statement on line 122: "the top WO_{3-x} layer exhibits nearly the same atomic thickness as that of monolayer WSe₂." seems inconsistent with the formation of nanowires. Shouldn't the nanowire profiles be evident after treatment? In general, I found the topological analysis of the nanowires to be lacking. Can the lengths and diameters of the wires be quantified? What are the average values? What variation is observed across different wires in a single sample, or for wires from different samples? Such statistical analysis seems basic and relatively easy to provide, therefore its absence is surprising.
- 2) The dashed lines in Fig 1d and 1e do not seem to match the AFM profiles in 1f. If the dashed lines are not related to the height profiles, then what information are they meant to convey?
- 3) The statement on line 113: "As the plasma treatment durations increase, the diameters and lengths of WO_x nanowires gradually decrease." is difficult to evaluate because of some inconsistencies in Fig S7. For example, it's not clear which images are magnified insets: Is b3 an inset of b2? Are d3 and d2 in different order from the others for some reason? This needs better explanation.
- 4) There seems to be a mistake in the scale bars of Fig S7. Are c1 and d1 on 20 μm scale? Should the scale in d2 and d3 be different? This sloppiness reduces confidence in the interpretation.
- 5) Strain mapping is unfamiliar to me, I did not understand the information obtained from this analysis based on the limited explanation provided.
- 7) Strong adsorption of MB to the WO_x sample seems to provide excellent reproducibility. While this is a nice result, it relies on "homogeneous adsorption" which therefore suggests that the effect may not be universal. Can the authors comment on the generality of this approach to other analytes? There is a brief comparison with other dye molecules, but overall the discussion on this point seems underdeveloped and therefore limits the impact of this work in terms of providing a versatile SERS substrate.
- 8) On line 209, I think it would be more accurate to say that the adsorption energy is "larger" or "stronger", rather than "obviously smaller" for WO_x compared with WSe₂. The point is that MB binds more strongly to the former, not that the stabilization energy is more negative.
- 9) In any event, the adsorption energy of 4.82 eV is a bit deceiving. This is the calculated energy in the absence of solvent polarization and other moderating effects. While the comparison between the two interfaces may have some value, the calculated value (~460 kJ/mol, equivalent to a covalent bond) is not the same as the actual binding energy in the real sample.
- 10) What is the explanation for the reduced SERS signal for MB in the mixed solution (Fig S35)? Is this a consequence of competitive binding by R6G, which is in much higher concentration?

11) I don't find this statement on lines 240-242 to be very convincing: "Different dye molecules have different LODs on WO_{3-x}/WSe₂ heterostructure substrates due to their different band alignments." Rather than different band alignments, isn't it more likely that MB simply binds stronger? Stronger binding seems to be the upshot from the different measurements.

12) The description of photo induced charge transfer (PICT) needs clarification: "The photo- induced charge transfer (PICT) transition from WO_{3-x} to MB could be beneficial to the enhancement of the SERS effect." Do the authors mean to suggest that this is an intermolecular charge-transfer transition? That would be reasonable, but I don't know how else the charge transfer might result in Raman enhancement.

13) The transient transmission results (Fig 5) seem to indicate CT from the surface to MB (likely instantaneous) followed by relaxation within ~1 ps. Other than that, I'm not clear what information can be extracted from the transient measurements.

13) Claims that "the absorption peak of MB/WO_x/WSe₂ is much higher..." (line 250) are exaggerated. In fact, the transmission for this sample (red line in Fig 4c) is clearly offset from the others, most likely due to increased scattering or other effects. This should not be confused with stronger absorption. Moreover, the absorption strength also depends on the number of molecules in the field of irradiation, which is not necessarily constant for the different samples (and, indeed, explains the negligible absorption in the case of the quartz surface, which does not adsorb MB).

14) Line 322: It is NOT accurate to say that this results is "reaching almost single molecule level". Even if the signal is observed at low concentration of the solution, strong binding would allow accumulation of the target dye molecule at the interface. This is very different from "single molecule" sensitivity.

Minor points:

The Raman laser wavelength (633 nm) should be reported sooner, for example near the beginning of the discussion on line 181.

Lines 306-307: have NG and PG been defined in the main text?

Point-by-point response to the reviewers' comments

Reviewer #1 (Remarks to the Author):

Developing efficient noble-metal-free SERS substrates for ultrasensitive molecular sensing is very meaningful but also very challenging. The authors report the use of a 1D/2D WO_{3-x}/WSe₂ heterostructure as highly-active SERS substrate, which achieves very high sensitivity for MB detection with a record low detection concentration of 5×10⁻¹⁸ M. The experimental results are interesting, and well presented with clear and useful figures. However, some revisions are necessary before the publication of the paper as highlighted below.

Comment #1-1) It seems that very high sensitivity of 10⁻¹⁸ M is only achieved with MB detection. How about other analytes? Why MB is so special? Considering the need for practical applications, it is better to detect a wide variety of analytes using 1D/2D WO_{3-x}/WSe₂ heterostructure, especially for more analytes in practical uses.

Response: We appreciate the referee's constructive comment very much. We have detected other molecules (e.g., R6G and CV) and discussed the results in details on Page 11 of the revised manuscript as below: "*The SERS effect of WO_{3-x}/WSe₂ heterostructure can also be extended to detect other dye molecules (e.g., rhodamine 6G (R6G), crystal violet (CV)). The Raman fingerprints of R6G and CV molecules on WO_{3-x}/WSe₂ are clearly distinguished (Supplementary Fig. 41a-b), as marked by the symbols of "♣" and "♦", respectively. And, the LODs of WO_{3-x}/WSe₂ for the detection of R6G and CV are superior to most of the reported non-noble-metal substrates (Supplementary Table 1). Even compared with Au NPs substrates, WO_{3-x}/WSe₂ heterostructure still exhibits much higher sensitivity (Supplementary Fig. 41c-d). The different LODs for the detection of various dye molecules on WO_{3-x}/WSe₂ heterostructures might be due to the different binding energy and charge transfer process between the substrates and the probe molecules, and different photo-induced charge transfer (PICT) process from the band edges of substrates to the affinity levels of probe molecules. Such demonstrated universal SERS effect of WO_{3-x}/WSe₂ will endow it promising potentials in many fields, including food safety, chemical analysis, environmental monitoring, etc.*"

The ultrasensitive SERS effect of 1D/2D WO_{3-x}/WSe₂ heterostructure for the detection of MB molecules might be ascribed to the much stronger binding energy and efficient charge transfer between the heterostructure and MB molecules than those of other dye molecules. And, the photo-induced charge transfer (PICT) process between heterostructure and MB molecules depending on their band alignment can be beneficial to the enhancement of the SERS effect due to the charge transfer resonance, which can borrow intensity from the molecular transition and exciton transition through the Herzberg-Teller coupling constant ($h\nu_I$ and $h\nu_K$) to induce MB molecules much more polarized, consequently increasing the Raman scattering cross-section and further enhancing the SERS effect.

Considering the need for practical applications, we further use 1D/2D WO_{3-x}/WSe₂ heterostructure to detect bio-molecules, such as DNA Plasmid, and make comparisons the detection sensitivity between our heterostructure and noble-metal substrates. The results show that DNA Plasmid can be readily detected on the WO_{3-x}/WSe₂ without any additional SERS tag modifications (Figure R1a). And, the LOD of DNA plasmid on WO_{3-x}/WSe₂ substrate is about 10⁻¹⁰ M. As a comparison, there are no distinct Raman fingerprints of DNA plasmid on noble-metal based SERS substrates (e.g., Au nanoparticles) at 1187, 1295,

and 1465 cm^{-1} due to the strong fluorescence background (Figure R1b).

Figure S41. Raman spectra of 1D/2D $\text{WO}_{3-x}/\text{WSe}_2$ heterostructures and Au NPs for probing different molecules. (a) Rhodamine 6G (R6G) and (b) Crystal Violet (CV) on $\text{WO}_{3-x}/\text{WSe}_2$ heterostructures. (c) R6G, and (d) CV on Au NPs. The distinct Raman fingerprints of R6G and CV molecules (marked by “♣” and “◆”, respectively) demonstrate the universal sensing capability of $\text{WO}_{3-x}/\text{WSe}_2$ heterostructures for detecting various dye molecules.

Figure R1. Raman spectra of DNA Plasmid on (a) 1D/2D $\text{WO}_{3-x}/\text{WSe}_2$ heterostructures and (b) Au nanoparticles (Au NPs).

Comment #1-2) How about the selectivity? Can the SERS substrate show good selectivity in the mixed analyte systems with similar molecular structures?

Response: We appreciate the important comment of the referee very much. About the selectivity, we prepared the mixed solution of MB and R6G molecules, which was shown on Page 11 of the revised manuscript as below: “*Meanwhile, by choosing different laser lines, Raman fingerprints of individual MB and R6G in the mixed solution can be clearly detected on WO_{3-x}/WSe_2 (Supplementary Fig. 42), respectively, demonstrating the excellent selectivity of WO_{3-x}/WSe_2 substrate.*”

“*When excited by 633 nm Raman laser, only the Raman fingerprints of MB molecules in the mixed solution can be observed on WO_{3-x}/WSe_2 substrate. When using 532 nm Raman laser, both the Raman fingerprints of MB and R6G molecules can be excited, achieving the excellent selectivity detection, which benefits the practical applications of WO_{3-x}/WSe_2 heterostructure in the future. The selective detection capability can be explained using the molecular transition resonance, namely, the matched molecular wavelength with the excited Raman laser wavelength ($\lambda_{mol} \approx \lambda_L$), which can contribute to increasing the Raman scattering cross-section¹⁰. Thus, this Raman laser dependent behavior can be related to the affinity levels of different molecules. The highest occupied molecular orbital (HOMO) and the lowest unoccupied molecular orbital (LUMO) levels of MB (R6G) are located at -6.26 (-5.70) and -4.55 (-3.40) eV, respectively. So, the energy difference of MB (R6G) between HOMO and LUMO is 1.71 (2.30) eV. Thus, 633 nm Raman laser can only excite the fingerprints of MB molecules, while 532 nm Raman laser can excite both the fingerprints of MB and R6G molecules.*”

Figure S42. Raman spectra of MB and R6G in the mixed solution collected on 1D/2D WO_{3-x}/WSe_2 heterostructure excited by 633 and 532 nm Raman laser.

Comment #1-3) The authors have emphasized the ultrafast charge transfer of their SERS substrate for sensing. Can the authors compare the charge transfer rates of their SERS substrate and other noble-metal-free and noble-metal SERS substrates using ultrafast transient spectroscopy?

Response: About the charge transfer between our SERS substrates and other non-noble-metal substrates, we have compared our 1D/2D WO_{3-x}/WSe_2 heterostructure with other non-noble-metal substrates (e.g., 2D/2D $WO_{3-x}-NF/WSe_2$ heterostructure), which was shown on Page 14 of the revised manuscript as below:

“In addition, the transient transmission spectroscopy of the MB adsorbed on 2D/2D WO_{3-x} -NF/ WSe_2 was carried out for comparison (Supplementary Fig. 49). The MB bleaching signal on WO_{3-x} -NF/ WSe_2 is nearly an order of magnitude weaker than that on 1D/2D WO_{3-x} / WSe_2 , suggesting that the 1D/2D WO_{3-x} / WSe_2 heterostructures exhibits an enhanced charge transfer process by almost a factor of ten.”

The enhancement mechanism of noble-metal-free SERS substrates is mainly based on the chemical enhancement induced from the charge transfer between the substrates and the probe molecules. So, the ultrafast transient spectroscopy was carried out to verify and quantify the charge transfer between noble-metal-free substrates and probe molecules. While, the enhancement mechanism of noble-metal SERS substrates is mainly based on the electromagnetic enhancement, which originates from the surface plasmon resonance rather than the charge transfer.

Figure S49. Ultrafast transient transmission spectra of MB/WO_{3-x} -NF/ WSe_2 and the comparison with nanowire-based MB/WO_{3-x} / WSe_2 . (a) Transient transmission images of MB/WO_{3-x} -NF/ WSe_2 , by using different pump lights: 532 nm, 633 nm, and 750 nm. The time-dependent curves at 690 nm and 750 nm are in red and blue, respectively. (b) Wavelength-dependent data at 1.5 ps for both nanowire-based MB/WO_{3-x} / WSe_2 (red curves) and nanoflake-based MB/WO_{3-x} -NF/ WSe_2 (black curves). All curves are normalized to the A exciton peak of WSe_2 . The red arrows mark the MB bleaching signal.

Comment #1-4) The experimental details for SERS detection should be more detailed. The enhancement factor can be calculated. Commercial metal SERS substrates can be used for comparison.

Response: We appreciate the referee’s constructive suggestion very much. Accordingly, we have added the experimental details for SERS detection on Page 17 of the revised manuscript as below: “**SERS measurements.** MB, R6G, and CV molecules were dissolved in ethanol to form solutions with the initial concentration of 5×10^{-5} M. Then, the dye solutions with different concentrations ranging from 5×10^{-6} to 5×10^{-18} M can be prepared by sequential dilution process. Samples grown on SiO₂/Si substrates (0.5×0.5 cm²) were immersed into the solutions with different concentrations for ~30 min, respectively, followed by naturally drying. Then, the samples were rinsed with ethanol to remove the free molecules and dried with N₂ gas. All the Raman spectra were collected in a HORIBA LabRAM HR system with the $\times 50$ objective. The Raman excitation wavelength for probing MB is 633 nm but for probing R6G and CV is 532 nm. The laser power is below 1 mW to avoid the possible heating effect on samples. The integration time for MB, R6G, and CV with the concentrations from 5×10^{-5} to 5×10^{-14} M is 5 s, while for MB with the concentrations from 5×10^{-15} to 5×10^{-18} M is 40 s. The Raman spectra were collected with the same experimental parameters for the better comparison among different 2D materials and noble-metal substrates.”

The enhancement factor of 1D/2D WO_{3-x}/WSe₂ for the detection of MB molecules has been calculated on Page 10 of the revised manuscript and Supplementary Information as below: “The EF of WO_{3-x}/WSe₂ was calculated to be 5.0×10^{11} (bulk MB as reference, Supplementary Fig. 38).”

The enhancement factor (EF) can be calculated according to the following equations^{8,9}:

$$EF = (I_{SERS}/N_{SERS}) / (I_{bulk}/N_{bulk})$$

$$N_{SERS} = cV N_A A_1 / A_{sub}$$

$$N_{bulk} = \rho h N_A A_1 / M$$

Where I_{SERS} and I_{bulk} are the Raman intensities of probe molecules on SERS substrate and bulk molecules on bare substrate, respectively. N_{SERS} and N_{bulk} refer to the amounts of probe molecules in the SERS test and in the bulk Raman test, respectively. V and c are the volume and concentration of probe molecules, respectively. M and ρ are the molar mass and density of probe molecules (319.86 g mol⁻¹, 1.757 g cm⁻³ for MB), respectively. A_1 is the illumination area of Raman laser. The layer penetration depth (h) can be calculated by the equation of $h = 2\lambda/NA^2$, in which NA is the numerical aperture of 0.5. The Raman intensity of MB on WO_{3-x}/WSe₂ with 40 s integration time is 393 counts and for bulk MB with 1 s integration time is 1374 counts (Fig. S38). The number of bulk MB molecules can be calculated using the density of bulk MB due to the high concentration of MB solution (5×10^{-2} M). Thus, the EF of MB on WO_{3-x}/WSe₂ can be calculated as:

$$EF = \frac{I_{SERS}}{I_{bulk}} \times \frac{\rho h A_{sub}}{cVM} = \frac{393/40}{1374/1} \times \frac{1.757 \times 5.064 \times 10^{-4} \times 0.25}{5 \times 10^{-18} \times 2 \times 10^{-3} \times 319.86} \approx 5.0 \times 10^{11}$$

Figure S38. Raman spectra of bulk MB with the concentration of 5×10^{-2} M on bare SiO_2/Si substrate.

We further measure the SERS performance of commercial noble-metal SERS substrates (Au NPs from Shandong Wise Sensing Technology Co. Ltd), as shown in Figure R2. It can be seen that the Raman peaks of MB molecules at 1620 cm^{-1} on the commercial Au NPs are hardly to be detected when the concentration of MB solution was decreased to 5×10^{-10} M, which is far inferior to that of 1D/2D $\text{WO}_{3-x}/\text{WSe}_2$ heterostructure.

Figure R2. Raman spectra of MB molecules with different concentrations on commercial Au NPs.

Reviewer #2 (Remarks to the Author):

Qian Lv et al have constructed a 1D nanowire pattern/2D flake heterostructures in a simple yet effective way, systematically studied the charge transfer effects and the correlation to the SERS activities. The heterostructures were well characterized and physical properties extensively investigated experimentally the theoretically. An extremely high SERS enhancement were achieved in the MB/1D WO_{3-x} /2D WSe_2 heterostructures. They have also uncovered the timescale of around 1.0 picosecond for the charge transfer process with ultrafast transient spectroscopy. It is a quite comprehensive study on charge transfer and SERS effects in this heterostructure. However, it lacks enough novelty nor is of general interest as required by Nature Communications. Some comments and suggestions:

Response: Thank you very much for your important comments. We appreciate the reviewer by pointing out that our work is “*a quite comprehensive study*”. Regarding the novelty, to the best of our knowledge, this is indeed the first report of using 1D/2D heterostructures as SERS substrates with ultrasensitive molecular sensing performance, achieving attomolar-level record. In details, first, we propose a fast, effective, and novel strategy to realize the synthesis of 1D/2D WO_{3-x} / WSe_2 heterostructure through the oxygen plasma treatment. Currently, TMDCs-based mixed-dimensional heterostructures are mainly synthesized by vapor growth, aligned transfer, electron beam, etc. which need complex and precise procedures, as well as exhibit poor scalability. Thus, developing an efficient and facile route to synthesize TMDCs-based heterostructures is of great significance for their practical applications but still challenging. Second, for the enhanced mechanism of the ultrahigh sensitivity of heterostructure, we carried out time-resolved transient absorption/transmission spectroscopy to investigate and quantify the charge transfer process between heterostructure substrates and probe molecules in experiments combined with DFT, not like previously reported studies only based on theory results, which make the enhancement mechanism of SERS substrates much clearer and more convincing and provide guidance for the further SERS mechanism study.

We note that the third reviewer also recognize our preparing strategy by writing that “The paper is well written and describes an interesting and promising approach for developing reproducible SERS substrates” (please see more details in the third reviewer’s comments). Moreover, this system also exhibits excellent universal molecular sensing capability not only for dye molecules, but also for biomolecules (e.g., DNA). These features could be exploited for widespread applications, including Raman sensing in food safety, chemical analysis, environmental monitoring, etc., which will excite great interest in a wide audience from nanotechnology, materials science, environmental science based on above-mentioned key innovations.

Comment #2-1) The manuscript is entitled “Ultrafast charge transfer of 1D nanowire pattern/2D flake heterostructures for attomolar-level molecular sensing”. However, there is no decisive correlation between the ultrafast charge transfer and attomolar-level molecular sensing. Is ~ 1 ps transfer time a necessary condition for such ultrahigh SERS sensitivity? What is the physics behind? Do other heterostructures with longer charge transfer time have to be with lower SERS enhancement factor?

Response: The SERS effect of 2D materials is usually attributed to the chemical mechanism, induced from the charge transfer between the substrates and the probe molecules. Our observation of the charge transfer time ~ 1 ps, together with our calculation, obviously support such mechanism, i.e., charge transfer between the heterostructure and the probe molecules. As we have emphasized in the manuscript, the efficient charge transfer is crucial for ultrasensitive SERS, benefiting from both of the unique structures of 1D WO_{3-x} nanowires and the effective interlayer coupling of heterostructure. As a comparison, for 2D/2D WO_{3-x} -

NF/WSe₂ heterostructures, due to the weak charge transfer process, it was difficult to detect the pump-induced bleaching signal of MB molecules in MB/WO_{3-x}-NF/WSe₂ system (marked by red arrows in Figure S49), which makes the measurement of the charge transfer timescale a big challenge. So, it is hardly to define that the longer charge transfer time means the lower SERS enhancement factor.

Figure S49. Ultrafast transient transmission spectra of MB/WO_{3-x}-NF/WSe₂ and the comparison with nanowire-based MB/WO_{3-x}/WSe₂. (a) Transient transmission images of MB/WO_{3-x}-NF/WSe₂, by using different pump lights: 532 nm, 633 nm, and 750 nm. The time-dependent curves at 690 nm and 750 nm are in red and blue, respectively. (b) Wavelength-dependent data at 1.5 ps for both nanowire-based MB/WO_{3-x}/WSe₂ (red curves) and nanoflake-based MB/WO_{3-x}-NF/WSe₂ (black curves). All curves are normalized to the A exciton peak of WSe₂. The red arrows mark the MB bleaching signal.

Comment #2-2) The authors attribute the strong SERS enhancement of the system to the combined influence of interfacial lattice distortion/strain, efficient electron transport along the axis directions and the existence of oxygen vacancies mediated vibronic coupling, which have already been reported in the literature. No new insight into the enhancement mechanism.

Response: We thank the referee for the constructive comment. For the individual WO_{3-x} nanowires, the SERS enhancement mechanism can be attributed to the three factors mentioned by the referee. However,

after constructing 1D/2D WO_{3-x}/WSe₂ heterostructure, the interlayer coupling of the heterostructure plays a significant role in further enhancing the SERS effect of the substrate, in which a charge transfer from the conduction band minimum (CBM) of WSe₂ to the CBM of WO_{3-x} occurs. The interlayer coupling of the heterostructure combined with the three factors mentioned above synergistically enhance the SERS effect of heterostructure and further achieve attomolar-level detection capability. DFT calculations also demonstrate that this charge transfer of heterostructure can further increase the adsorption energy and increase the charge transfer between MB and WO_{3-x}/WSe₂ to improve the probe sensitivity.

Comment #2-3) The SERS measurements showed that charge transfer between MB and WO_{3-x}/WSe₂ is laser line dependent (Figure S39). Fig 5c indicates that under the resonance effect the charge transfer time scale is about 0.5 ps, whereas the off-resonance charge transfer time scale 1.0 ps, which appears to be caused by a number of effects outside the scope of this manuscript.

Response: In the revised Supplementary Information, Figure S43 (Figure S39 in the last version) shows the Raman spectra of MB excited by different laser lines, hardly provide charge transfer time constant. In Fig. 5c, ~0.5 ps is the thermalization time for overgap pumping by 532 and 633 nm, and ~1 ps is the charge transfer time from WSe₂ to MB assisted by WO_{3-x} for resonant excitation of WSe₂ A-exciton at 750 nm.

In ultrafast physics, when pump energy is above the bandgap, the initially excited hot electrons with energy higher than the bandgap, will rapidly lose energy through electron-electron and electron-phonon scattering and accumulate around the bottom of conduction bands. This is well-known as the **thermalization process** (timescale can be from femtoseconds to picoseconds) that allows detection of the bleaching signal at the bandgap energy by the second probe light. In Fig. 5c, short wavelengths (532 and 633 nm) induce overgap excitation of both MB (bandgap at ~690 nm) and WSe₂ (A-excitation at ~750 nm). Then thermalization process gives rise to bleaching maximums at ~ 0.5 ps for MB (time-dependent curve at 690 nm) and WSe₂ (time-dependent curve at 750 nm) after pump light off.

When pump light is tuned to 750 nm, that is resonant excitation of WSe₂ A-excitation only, and well below the bandgap of MB. Therefore, the thermalization process does not occur for MB and the 0.5 ps bleaching maximum becomes absent in the MB curve. Instead, a slower slop change/peak in MB curve emerges about 1 ps after the WSe₂ bleaching peak (0.2 - 0.3 ps determined by the overall temporal resolution), indicating a charge transfer from WSe₂ to MB.

In order to explain the ultrafast effects more clearly, we have modified on Page 14 of the revised manuscript as below: "*nearly identical to that of the WSe₂ A exciton peak due to the thermalization process, i.e., high-energy hot carriers relax to the bottom of conduction bands*". Thank you so much!

Comment #2-4) The authors emphasize the strain effects on charge transfer and thus SERS enhancement in WO_{3-x}/WSe₂. It'd be good that the authors can give the value of the measured strain that might be correlated to the interfacial distance between adjacent layers and thus charge redistribution as revealed by DFT calculation.

Response: Thank you for this important comment and advice. The strain at the interfaces of WO_{3-x} nanowires has been measured based on GPA images on Page 18 of the revised Supplementary Information as below: "*In terms of the WO_{3-x} nanowires aligned along the three-fold symmetric directions of the WSe₂ (Fig. S17a), there exist strain at the interfaces of WO_{3-x} nanowires due to the different crystallographic*

orientation. Thus, we identify and quantify the strain at the nanowire interfaces by using geometric phase analysis (GPA) method. It was carried out by the free FRWRtools plugin (see the link shown below) for Digital Micrograph software. Here, the strain field distributions were illustrated by the lattice fringes variations of the HRTEM image across the fields of view. To map the strain field distribution, the unstrained area of nanowire II was chosen as the reference lattice with the direction x and y corresponding to $[200]$ and $[020]$ (Fig. S17b), respectively. Thus, only the strain field distribution in the region related to nanowire II can be analyzed, which include I-II and III-II nanowire interfaces. Note that the GPA color maps on nanowire I and III based on the nanowire II unstrained area as reference lattice can not be used as the real evidence to reflect the strain field distribution. Fig. S17c shows the ϵ_{xx} strain map, indicating the existence of x -direction normal strain at the interface between II and III nanowires. The ϵ_{yy} map in Fig. S17d shows the y -direction normal strain, which is mainly distributed at the I-II and III-II nanowires interfaces. The ϵ_{xy} map in Fig. S17e represents the shear strain, and the ϵ_{xy} is also mainly distributed at the interfaces of the I-II and III-II nanowires. According to the strain intensity profile (Fig. S17f), the strain at the interfaces of WO_{3-x} nanowires is $\sim 1\%$.

Link for FRWRtools plugin: www.physics.hu-berlin.de/en/sem/software/software_frwrtools

Figure S17. Strain-field mapping at the interfaces of WO_{3-x} nanowires. (a) HRTEM image of three intersected WO_{3-x} nanowires. (b) A superimposed image and its ϵ_{yy} strain map. (c) ϵ_{xx} , (d) ϵ_{yy} and (e) ϵ_{xy} strain maps. (f) The corresponding strain intensity profile at the interfaces of WO_{3-x} nanowires along the white dotted line in (c).

Comment #2-5) The manuscript showed nice AFM images of WO_{3-x}/WSe_2 heterostructures. It'd be good if the authors can also present the AFM line scan profiles across the WO_{3-x} nanowires on WSe_2 , which can clearly indicate the height of nanowires and 1D/2D stacking morphology.

Response: The AFM line scan profile across the WO_{3-x} nanowires on WSe_2 has been added on Page 6 of the revised manuscript as below: “The AFM profile indicates that the thickness of an individual WO_{3-x} nanowire is nearly 2.0 nm (Fig. 1f).” Thank you!

Fig. 1 Formation and morphology of 1D/2D WO_{3-x}/WSe₂ heterostructures. *a* Schematic illustration of the evolution of 1D oriented WO_{3-x} nanowires from 2D WSe₂ flake via oxygen plasma treatment. *b-c* Optical images of pristine WSe₂ and WO_{3-x}/WSe₂ heterostructures. *d-e* Atomic force microscope (AFM) images of pristine WSe₂ and WO_{3-x}/WSe₂ heterostructures. *f* Height profile of an individual WO_{3-x} nanowire corresponding to the white dotted line in (*e*). *g* Raman spectra, *h* photoluminescence spectra, and *i* X-ray photoelectron spectroscopy (XPS) fine scan spectra of pristine WSe₂ and WO_{3-x}/WSe₂ heterostructures.

Reviewer #3 (Remarks to the Author):

This manuscript describes the fabrication of WO_x nanowires on top of few-layer flakes of WSe_2 as a new medium for SERS spectroscopy. The fabrication method is reasonably straightforward and has been characterized with several imaging and spectroscopic methods. The authors then demonstrate SERS activity for the dye molecule methylene blue (MB). They claim that the WO_x nanowires enable an extremely low limit of detection. Finally, the authors use ultrafast spectroscopy in an attempt to probe the mechanism of the SERS enhancement.

Overall, the paper is well written and describes an interesting and promising approach for developing reproducible SERS substrates. In fact, the apparent reproducibility in making the wires (e.g., Fig S2) and especially the apparent reproducibility of the SERS intensity for different samples (Fig 3c) is intriguing. The manuscript has a few inconsistencies and other deficiencies that limit the impact of the work, but these are addressable and a revised version of the manuscript may be suitable for publication so that other researchers can be made aware of the work.

At a minimum, the authors need to address the following concerns (listed in no particular order).

Comment #3-1) The statement on line 122: “the top WO_{3-x} layer exhibits nearly the same atomic thickness as that of monolayer WSe_2 .” seems inconsistent with the formation of nanowires. Shouldn't the nanowire profiles be evident after treatment? In general, I found the topological analysis of the nanowires to be lacking. Can the lengths and diameters of the wires be quantified? What are the average values? What variation is observed across different wires in a single sample, or for wires from different samples? Such statistical analysis seems basic and relatively easy to provide, therefore its absence is surprising.

Response: We appreciate the constructive comment from the referee very much. We apologize for the statement on line 122. We have carefully checked the original AFM data, measured the nanowire profile and modified the statement. And, the related AFM profile of nanowire has been supplied on Page 6 of the revised manuscript as below: “*The AFM profile indicates that the thickness of an individual WO_{3-x} nanowire is nearly 2.0 nm (Fig. 1f).*”

Also, the statistic length and diameter distributions of WO_{3-x} nanowires as the function of the plasma treatment durations have been added on Page 5 of the revised manuscript as below: “*According to the statistic length and diameter distributions of WO_{3-x} nanowires as the function of the plasma treatment durations (Supplementary Figs. 8-9), the lengths and diameters of WO_{3-x} nanowires gradually decrease with increasing the treatment durations.*”

Fig. 1 Formation and morphology of 1D/2D WO_{3-x}/WSe₂ heterostructures. *a* Schematic illustration of the evolution of 1D oriented WO_{3-x} nanowires from 2D WSe₂ flake via oxygen plasma treatment. *b-c* Optical images of pristine WSe₂ and WO_{3-x}/WSe₂ heterostructures. *d-e* Atomic force microscope (AFM) images of pristine WSe₂ and WO_{3-x}/WSe₂ heterostructures. *f* Height profile of an individual WO_{3-x} nanowire corresponding to the white dotted line in (*e*). *g* Raman spectra, *h* photoluminescence spectra, and *i* X-ray photoelectron spectroscopy (XPS) fine scan spectra of pristine WSe₂ and WO_{3-x}/WSe₂ heterostructures.

Figure S8. Statistic length (L) distributions of WO_{3-x} nanowires as the function of the plasma treatment durations. (a) Plasma 15s. (b) Plasma 30 s. (c) Plasma 45 s. (d) Plasma 60 s. It can be seen that with prolonging the plasma durations, the lengths of WO_{3-x} nanowires gradually decrease.

Figure S9. Statistic diameter (D) distributions of WO_{3-x} nanowires as the function of the plasma treatment durations. (a) Plasma 15s. (b) Plasma 30 s. (c) Plasma 45 s. (d) Plasma 60 s. It can be seen that with prolonging the plasma durations, the diameters of WO_{3-x} nanowires gradually decrease.

Comment #3-2) The dashed lines in Fig 1d and 1e do not seem to match the AFM profiles in 1f. If the dashed lines are not related to the height profiles, then what information are they meant to convey?

Response: The dashed line and AFM profile have been modified on Page 6 of the revised manuscript as below: “*The AFM profile indicates that the thickness of an individual WO_{3-x} nanowire is ~ 2.0 nm (Fig. 1f).*” We thank the referee’s comment once again.

Comment #3-3) The statement on line 113: “As the plasma treatment durations increase, the diameters and lengths of WO_x nanowires gradually decrease.” is difficult to evaluate because of some inconsistencies in Fig S7. For example, it’s not clear which images are magnified insets: Is b3 an inset of b2? Are d3 and d2 in different order from the others for some reason? This needs better explanation.

Response: Many thanks for this comment. To make it clear, we have rewritten the caption of Fig. S7. In Fig. S7, b₃ is an inset of b₂. And, d₃ and d₂ have been exchanged in order. Based on the new order as below, d₂ is an inset of d₁.

Figure S7. Atomic force microscope (AFM) images of WO_{3-x}/WSe_2 heterostructures with different plasma durations. (a_1 - a_3) plasma 15 s. (b_1 - b_3) plasma 30 s. (c_1 - c_3) plasma 45 s. (d_1 - d_3) plasma 60 s. The a_2 and a_3 are zoom-in images of a_1 , b_2 and b_3 are zoom-in images of b_1 , c_2 is zoom-in of c_1 , and d_2 is zoom-in image of d_1 .

Comment #3-4) There seems to be a mistake in the scale bars of Fig S7. Are c_1 and d_1 on 20 μm scale? Should the scale in d_2 and d_3 be different? This sloppiness reduces confidence in the interpretation.

Response: We are very grateful to the referee for pointing this out. In fact, the CVD-grown WSe_2 flakes exhibit different size. In Fig. 7a₁, the initial WSe_2 flake is large and so the scale bar is 20 μm . While, the initial WSe_2 flakes in Fig. S7c₁ and d₁ are relatively small compared to that in Fig. S7a₁, leading to the much smaller scale bars. Here, we present the raw AFM images of Fig. 7a₁, c₁, and d₁ in order to show their size difference (Figure R3). Moreover, the scale bar in Fig. S7d₁ has been modified. And, the scale bars in d₂ and d₃ are different, because the d₂ and d₃ are from different areas. The longer WO_{3-x} nanowires in d₃ might be formed at the initial stage of plasma treatment, which account for small fraction.

Figure R3. Raw AFM images of WO_{3-x}/WSe_2 heterostructures. AFM images of (a), (b) and (c) are corresponding to that of Fig. S7a₁, c₁ and d₁, respectively.

Comment #3-5) Strain mapping is unfamiliar to me, I did not understand the information obtained from this analysis based on the limited explanation provided.

Response: We thank the important comment of the referee. The explanation of strain mapping in details has been provided on Page 18 of the revised Supplementary Information as below: “*In terms of the WO_{3-x} nanowires aligned along the three-fold symmetric directions of the WSe_2 (Fig. S17a), there exist strain at the interfaces of WO_{3-x} nanowires due to the different crystallographic orientation. Thus, we identify and quantify the strain at nanowire interfaces by using geometric phase analysis (GPA) method. It was carried out by the free FRWRtools plugin (see the link shown below) for Digital Micrograph software. Here, the strain field distributions were illustrated by the lattice fringes variations of the HRTEM image across the fields of view. To map the strain field distribution, the unstrained area of nanowire II was chosen as the reference lattice with the direction x and y corresponding to $[200]$ and $[020]$ (Fig. S17b), respectively. Thus, only the strain field distribution in the region related to nanowire II can be analyzed, which include I-II and III-II nanowire interfaces. Note that the GPA color maps on nanowire I and III based on the nanowire II unstrained area as reference lattice can not be used as the real evidence to reflect the strain field distribution. Fig. S17c shows the ϵ_{xx} strain map, indicating the existence of x -direction normal strain at the interface between II and III nanowires. The ϵ_{yy} map in Fig. S17d shows the y -direction normal strain, which is mainly distributed at the I-II and III-II nanowires interfaces. The ϵ_{xy} map in Fig. S17e represents the shear strain, and the ϵ_{xy} is also mainly distributed at the interfaces of the I-II and III-II nanowires. According to the strain intensity profile (Fig. S17f), the strain at the interfaces of WO_{3-x} nanowires is $\sim 1\%$.*”

Link for FRWRtools plugin: www.physics.hu-berlin.de/en/sem/software/software_frwrtools”

Figure S17. Strain-field mapping at the interfaces of WO_{3-x} nanowires. (a) HRTEM image of three intersected WO_{3-x} nanowires. (b) A superimposed image and its ϵ_{yy} strain map. (c) ϵ_{xx} , (d) ϵ_{yy} and (e) ϵ_{xy} strain maps. (f) The corresponding strain intensity profile at the interfaces of WO_{3-x} nanowires along the white dotted line in (c).

Comment #3-6) Strong adsorption of MB to the WO_x sample seems to provide excellent reproducibility. While this is a nice result, it relies on “homogeneous adsorption” which therefore suggests that the effect may not be universal. Can the authors comment on the generality of this approach to other analytes? There is a brief comparison with other dye molecules, but overall the discussion on this point seems underdeveloped and therefore limits the impact of this work in terms of providing a versatile SERS substrate.

Response: Regarding the generality of WO_{3-x}/WSe_2 for the detection of other analytes, we have further discussed it in detail on Page 11 of the revised manuscript as below: “*The SERS effect of WO_{3-x}/WSe_2* ”

heterostructure can also be extended to detect other dye molecules (e.g., rhodamine 6G (R6G), crystal violet (CV)). The Raman fingerprints of R6G and CV molecules on WO_{3-x}/WSe_2 are clearly distinguished (Supplementary Fig. 41a-b), as marked by the symbols of “♣” and “◆”, respectively. And, the LODs of WO_{3-x}/WSe_2 for the detection of R6G and CV are superior to most of the reported non-noble-metal substrates (Supplementary Table 1). Even compared with Au NPs substrates, WO_{3-x}/WSe_2 heterostructure still exhibits much higher sensitivity (Supplementary Fig. 41c-d). The different LODs for the detection of various dye molecules on WO_{3-x}/WSe_2 heterostructures might be due to the different binding energy and charge transfer process between the substrates and the probe molecules, and different photo-induced charge transfer (PICT) process from the band edges of substrates to the affinity levels of probe molecules. Such demonstrated universal SERS effect of WO_{3-x}/WSe_2 will endow it promising potentials in many fields, including food safety, chemical analysis, environmental monitoring, etc.”

Figure S41. Raman spectra of 1D/2D WO_{3-x}/WSe_2 heterostructures and Au NPs for probing different molecules. (a) Rhodamine 6G (R6G) and (b) Crystal Violet (CV) on WO_{3-x}/WSe_2 heterostructures. (c) R6G, and (d) CV on Au NPs. The distinct Raman fingerprints of R6G and CV molecules (marked by “♣” and “◆”, respectively) demonstrate the universal sensing capability of WO_{3-x}/WSe_2 heterostructures for detecting various dye molecules.

Considering the need for practical applications, we further use 1D/2D WO_{3-x}/WSe_2 heterostructure to detect bio-molecules, such as DNA Plasmid, and make comparisons the detection sensitivity between our heterostructures and noble-metal substrates. The results show that DNA Plasmid can be readily detected on

the $\text{WO}_{3-x}/\text{WSe}_2$ without any additional SERS tag modifications (Figure R1a). The LOD of DNA plasmid on $\text{WO}_{3-x}/\text{WSe}_2$ substrate is about 10^{-10} M. As a comparison, there are no distinct Raman fingerprints of DNA plasmid on noble-metal based SERS substrates (e.g., Au nanoparticles) at 1187, 1295, and 1465 cm^{-1} due to the strong fluorescence background (Figure R1b).

Figure R1. Raman spectra of DNA Plasmid on (a) 1D/2D $\text{WO}_{3-x}/\text{WSe}_2$ heterostructures and (b) Au nanoparticles (Au NPs).

Comment #3-7) On line 209, I think it would be more accurate to say that the adsorption energy is “larger” or “stronger”, rather than “obviously smaller” for WO_x compared with WSe_2 . The point is that MB binds more strongly to the former, not that the stabilization energy is more negative.

Response: We thank the referee for the nice advice. Accordingly, the related expression has been modified on Page 9 of the revised manuscript as below: “*the adsorption energy for MB molecule on the WO_{3-x} substrate is -4.82 eV, which is obviously stronger than that on the WSe_2 substrate (-1.63 eV).*”

Comment #3-8) In any event, the adsorption energy of 4.82 eV is a bit deceiving. This is the calculated energy in the absence of solvent polarization and other moderating effects. While the comparison between the two interfaces may have some value, the calculated value (~ 460 kJ/mol, equivalent to a covalent bond) is not the same as the actual binding energy in the real sample.

Response: We thank the reviewer for this valuable comment. For the MB/ WSe_2 , the minimum distance between the N atom in MB and the Se atom in WSe_2 is 3.42 Å, suggesting the van der Waals interaction between MB and WSe_2 . While for the MB/ WO_{3-x} , the minimum distance between the N atom in MB and the W atom in WO_{3-x} is 2.53 Å, which is slightly larger than the N-W bond lengths in $h\text{-W}_2\text{N}_3$, $r\text{-W}_2\text{N}_3$, $c\text{-W}_3\text{N}_4$ and $\delta\text{-WN}$ (from 2.06 Å to 2.19 Å) (*Chem. Mater.*, **2012**, 24, 3023-3028). The N-W distance in the MB/ WO_{3-x} , which is much smaller than that of the van der Waals interaction and slightly larger than that of the N-W covalent interaction, indicates the existence of a strong interaction between the MB molecule and the WO_{3-x} substrate. In addition, the adsorbed MB has a large structural deformation on the WO_{3-x} substrate, while it is hardly distorted on the WSe_2 substrate. In contrast to the layered WSe_2 , the WO_{3-x} substrate has a three-dimensional structure with surface O defects, which is favorable for the adsorption of the molecule. Furthermore, the values of adsorption energy for MB/2H-NbS₂ and MB/1T-PtSe₂ are -3.51 and -3.76 eV, respectively, indicating that MB has a strong binding to the substrates, as shown in **Table R1**. Therefore, we believe that the adsorption energy of 4.82 eV in MB/ WO_{3-x} is reasonable.

Table R1 Adsorption energy (E_{ads}) of MB on various 2D materials

System	E_{ads} (eV)	Reference
MB/graphene	-1.64	
MB/1T-MoS ₂	-3.42	ACS Nano , 2019 , 13, 8312-8319
MB/2H-NbS ₂	-3.51	
MB/1T-PtSe ₂	-3.76	
MB/1T'-ReS ₂	-1.95	Nanoscale Horiz. , 2021 , 6, 186-191

Comment #3-9) What is the explanation for the reduced SERS signal for MB in the mixed solution (Fig S35)? Is this a consequence of competitive binding by R6G, which is in much higher concentration?

Response: We agree with the referee about this. The Raman intensity of MB in the mixed solution was weaker than that in the pure solution, which might be attributed to the competitive absorption of R6G molecules on the WO_{3-x}/WSe₂ heterostructure. The absorption of R6G molecules can lead to the reduced absorption sites of MB molecules on the WO_{3-x}/WSe₂ heterostructure, resulting in the weaker SERS signal of MB molecules in the mixed solution.

Comment #3-10) I don't find this statement on lines 240-242 to be very convincing: "Different dye molecules have different LODs on WO_{3-x}/WSe₂ heterostructure substrates due to their different band alignments. Rather than different band alignments, isn't it more likely that MB simply binds stronger? Stronger binding seems to be the upshot from the different measurements.

Response: We appreciate the advice from the referee and we have modified the statement on Page 11 of the revised manuscript as below: "*The different LODs for the detection of various dye molecules on WO_{3-x}/WSe₂ heterostructures might be due to the different binding energy and charge transfer process between the substrates and the probe molecules, and different photo-induced charge transfer (PICT) process from the band edges of substrates to the affinity levels of probe molecules.*"

Comment #3-11) The description of photo induced charge transfer (PICT) needs clarification: "The photo-induced charge transfer (PICT) transition from WO_{3-x} to MB could be beneficial to the enhancement of the SERS effect." Do the authors mean to suggest that this is an intermolecular charge-transfer transition? That would be reasonable, but I don't know how else the charge transfer might result in Raman enhancement.

Response: The enhancement mechanism of non-noble-metal SERS substrates is the chemical mechanism, which mainly originates from the charge transfer process between substrates and probe molecules. Thus, the efficient charge transfer process between the substrates and molecules can make the probe molecules much more polarized, leading to increasing the Raman scattering cross-section and further enhancing the Raman scattering.

Comment #3-12) The transient transmission results (Fig 5) seem to indicate CT from the surface to MB (likely instantaneous) followed by relaxation within ~1 ps. Other than that, I'm not clear what information can be extracted from the transient measurements.

Response: The enhancement mechanism of 2D SERS substrates is mainly based on the chemical enhancement induced from the charge transfer between the substrates and the probe molecules, namely, more efficient charge transfer could improve the sensitivity of SERS substrates. Thus, we use ultrafast transient spectroscopy to measure and compare the charge transfer process between various substrates and molecules. As shown in Figure 5a-b, due to the weak absorption and charge transfer, there are no pump-induced bleaching signals of MB on WSe₂. In contrast, the pump-induced bleaching signal of MB can be observed on WO_{3-x}/WSe₂ in terms of the strong absorption and efficient charge transfer between them, which benefits for the enhancement of Raman scattering and improves the detection sensitivity. Further, according to the Figure 5c-d, the charge transfer timescale between WO_{3-x}/WSe₂ and MB molecules is accurately quantified and around 1 ps. This is the first time to unveil the charge transfer process between SERS substrates and probe molecules using ultrafast transient spectroscopy, to the best of our knowledge, which make the enhancement mechanism of SERS substrates much clearer and more convincing and provide guidance for the further SERS mechanism study.

Fig. 5 Ultrafast transient absorption/transmission spectra of MB/WO_{3-x}/WSe₂. **a** Steady-state optical absorption spectra of MB/WO_{3-x}/WSe₂, MB/WSe₂, WO_{3-x}/WSe₂ and WSe₂. **b** Transient transmission curves of MB/WO_{3-x}/WSe₂ at a fixed delay time $t = 1.5$ ps with a pump wavelength of 532 nm. The dashed zero lines are shifted together with the corresponding data curves. **c** Transient transmission images of MB/WO_{3-x}/WSe₂ with different pump wavelengths of 532 nm, 633 nm, and 750 nm. The time-dependent curves at 690 nm and 750 nm are in red and blue, respectively. The black arrow depicts the charge transfer timescale of about 1.0 ps. **d-e** Secondary derivative images with respect to the wavelength (**d**) and the delay time (**e**), corresponding to the original data in (**c**). The gray solid curves in (**d**) are wavelength-dependent data at 1.5 ps, plotted with the dashed zero lines. Full comparisons are shown in Supplementary Fig. 47.

Comment #3-13) Claims that “the absorption peak of MB/ WO_x / WSe_2 is much higher...” (line 250) are exaggerated. In fact, the transmission for this sample (red line in Fig 4c) is clearly offset from the others, most likely due to increased scattering or other effects. This should not be confused with stronger absorption. Moreover, the absorption strength also depends on the number of molecules in the field of irradiation, which is not necessarily constant for the different samples (and, indeed, explains the negligible absorption in the case of the quartz surface, which does not adsorb MB).

Response: We appreciate the constructive comment of the referee and have modified the claims on Page 12 of the revised manuscript as below: “*In stark contrast, the absorption peak of MB molecules can be obviously observed upon absorbed on WO_{3-x} and $\text{WO}_{3-x}/\text{WSe}_2$. Meanwhile, the relative intensity and location of MB peaks on WO_{3-x} and $\text{WO}_{3-x}/\text{WSe}_2$ are remarkably changed compared to those of pristine MB, demonstrating the strong adsorption of MB onto WO_{3-x} and $\text{WO}_{3-x}/\text{WSe}_2$, which can further facilitate the efficient electron transition probability rate between them.*”

Comment #3-14) Line 322: It is NOT accurate to say that this results is “reaching almost single molecule level”. Even if the signal is observed at low concentration of the solution, strong binding would allow accumulation of the target dye molecule at the interface. This is very different from “single molecule” sensitivity.

Response: We appreciate the important comment of the referee. The related expression has been deleted on Page 15 of the revised manuscript. We thank the referee once again.

Minor points:

Comment #3-15) The Raman laser wavelength (633 nm) should be reported sooner, for example near the beginning of the discussion on line 181.

Response: We thank and agree with the referee. The Raman laser wavelength (633 nm) has been reported sooner on Page 8 of the revised manuscript as below: “*Figure 3a illustrates the Raman scattering process of MB molecules on $\text{WO}_{3-x}/\text{WSe}_2$ heterostructures excited by 633 nm laser.*”

Comment #3-16) Lines 306-307: have NG and PG been defined in the main text?

Response: We appreciate the comment of the referee very much. The definition of NG and PG has been supplied on Page 14 of the revised manuscript as below: “*Nitrogen-doped graphene (NG) presents a higher SERS effect than that of pristine graphene (PG), suggesting that defect engineering significantly influences the SERS effect.*”

REVIEWERS' COMMENTS

Reviewer #1 (Remarks to the Author):

Basically, the authors well revised the manuscript taking into account most of the review's comments. I have minor comments below: As for the charge transfer rates between SERS substrates and probe molecules, the authors may have better compare their 1D/2D WO₃-x/WSe₂ heterostructure with single WO₃-x, single WSe₂, single WO₃ and 1D/2D WO₃/WSe₂ heterostructure, highlighting the effectiveness of the interlayer coupling of the heterostructures.

Reviewer #2 (Remarks to the Author):

The authors have addressed my questions carefully and made necessary modifications accordingly. I recommend publishing the revised manuscript as it is.

Reviewer #3 (Remarks to the Author):

The authors have adequately addressed most of my concerns. It's unfortunate that some of their thoughtful responses didn't make it into the revised manuscript, as I think the manuscript could have been improved. For example, the response to comment #3-11 is helpful and I wish the authors had included this clear description (or a version of it) in the manuscript. Another example is the observation of SERS signal for the DNA plasmids. This is a nice example of the potential breadth of this approach, although I can understand that the authors did not want to add another (probably preliminary) result at this late stage. (Indeed, it's rather surprising that there is no hint of the SERS signal at 10⁻¹¹ M compared with the relatively robust signal at 10⁻¹⁰ M, so this result would probably need more verification before it could be included for publication.)

I'm also not sold on the explanation of the calculated 4.82 eV binding energy for MB/WO₃-x. This very large binding energy indicates a covalent interaction. Such a strong interaction may very well be the case, and would likely explain the large SERS response for MB. However, it further highlights the thorny issue of whether the analyte poisons the substrate. 4.82 eV binding energy is hardly reversible compared with kT of ~0.026 eV. Again, I think this may be the difference between calculated binding energies and the real interaction with solvent-screening. Nevertheless, there seems to be precedent in the literature to report these numbers and it is not a major point of the paper.

Overall, the fundamental science in the manuscript is suitable to publish despite some potential shortcomings related to the applications.

Point-by-point response to the reviewers' comments

Reviewer #1 (Remarks to the Author):

Basically, the authors well revised the manuscript taking into account most of the review's comments. I have minor comments below: As for the charge transfer rates between SERS substrates and probe molecules, the authors may have better compare their 1D/2D $\text{WO}_{3-x}/\text{WSe}_2$ heterostructure with single WO_{3-x} , single WSe_2 , single WO_3 and 1D/2D WO_3/WSe_2 heterostructure, highlighting the effectiveness of the interlayer coupling of the heterostructures.

Response: We thank the comment of the referee very much. The charge transfer between WSe_2 and MB is shown in the Supplementary Information Fig. 47 as below. Due to the weak charge transfer process, there is no a pump-induced bleaching signal of MB molecules on WSe_2 . The same as to WSe_2 , the pump-induced bleaching signal of MB molecules can not be detected on WO_{3-x} as shown in Fig. R1. So, it is hard to compare the charge transfer rate. In stark contrast, a pump-induced bleaching signal of MB molecules at 690 nm can be excited on 1D/2D $\text{WO}_{3-x}/\text{WSe}_2$ heterostructures with the charge transfer timescale at 1.0 ps, demonstrating the significant role of the interlayer coupling in enhancing SERS effect.

Figure S47. Ultrafast transient transmission images of MB/ $\text{WO}_{3-x}/\text{WSe}_2$, MB/ WSe_2 , $\text{WO}_{3-x}/\text{WSe}_2$ and WSe_2 , by using different pump lights: 532 nm (a), 633 nm (b), 750 nm (c and d). The right panels in (c) and (d) are second derivative images with respect to the wavelength.

Figure R1. Ultrafast transient transmission image of MB/WO_{3-x}.

Reviewer #2 (Remarks to the Author):

The authors have addressed my questions carefully and made necessary modifications accordingly. I recommend publishing the revised manuscript as it is.

Response: We appreciate the referee's important advice and positive comment very much!

Reviewer #3 (Remarks to the Author):

The authors have adequately addressed most of my concerns. It's unfortunate that some of their thoughtful responses didn't make it into the revised manuscript, as I think the manuscript could have been improved. For example, the response to comment #3-11 is helpful and I wish the authors had included this clear description (or a version of it) in the manuscript. Another example is the observation of SERS signal for the DNA plasmids. This is a nice example of the potential breadth of this approach, although I can understand that the authors did not want to add another (probably preliminary) result at this late stage. (Indeed, it's rather surprising that there is no hint of the SERS signal at 10^{-11} M compared with the relatively robust signal at 10^{-10} M, so this result would probably need more verification before it could be included for publication.)

Response: We thank the comments and advices of the referee. Accordingly, the responses to comment#3-11 have been added on Page 12 of the revised manuscript as below: “*the PICT transition (1.85 eV) from WO_{3-x} VBM to MB LUMO are beneficial to the enhancement of the SERS effect due to the charge transfer resonance, which can borrow intensity from the molecular transition and exciton transition through the Herzberg-Teller coupling constant (h_{VI} and h_{CK}) to make the probe molecules much more polarized and further increase the Raman scattering cross-section.*”

About the data of 1D/2D WO_{3-x}/WSe_2 heterostructures for the detection of DNA, as pointed out by the referee, it is only a preliminary result to verify that our heterostructures can not only detect various dye molecules, but also bio-molecules, suggesting the universal probability of 1D/2D WO_{3-x}/WSe_2 heterostructures for practical applications. We totally agree with the referee that “this result would probably need more verification before it could be included for publication.” Therefore, more measurement results and theory calculations will be carried out in the future to make the DNA detection of 1D/2D WO_{3-x}/WSe_2 heterostructures much more convinced and clearer. So, after comprehensive consideration, the data about the detection of DNA plasmids may be not suitable to be added into the manuscript at this stage. We thank you once again for this important advice.

I'm also not sold on the explanation of the calculated 4.82 eV binding energy for MB/ WO_{3-x} . This very large binding energy indicates a covalent interaction. Such a strong interaction may very well be the case, and would likely explain the large SERS response for MB. However, it further highlights the thorny issue of whether the analyte poisons the substrate. 4.82 eV binding energy is hardly reversible compared with kT of ~ 0.026 eV. Again, I think this may be the difference between calculated binding energies and the real interaction with solvent-screening. Nevertheless, there seems to be precedent in the literature to report these numbers and it is not a major point of the paper.

Response: We thank the reviewer for this valuable comment. The strength of ionic or covalent bond interactions is typically in the range of 1 eV-10 eV/atom, which is two to three orders of magnitude higher than van der Waals interactions, which are typically in the range of 1 meV-100 meV/atom. In our study, we have calculated that the binding energy between the MB molecule and the WO_{3-x} substrate is 4.82 eV, mainly due to van der Waals and dipole-dipole interactions. Furthermore, the MB molecule consists of 38 atoms, suggesting that the interaction between the MB molecule and the WO_{3-x} substrate is about 126 meV/atom, similar to that of two-dimensional materials such as MoS_2 (77 meV/atom) and SnSe (150 meV/atom) as reported in *Phys. Rev. Lett.* **118**, 106101(2017). Moreover, we agree with the reviewer's concern that the actual interaction between the MB molecule and the WO_{3-x} substrate may be affected by

the solvent screening. Given our results, we believe that a binding energy of 4.82 eV for the MB/WO_{3-x} interaction is reasonable and that the analyte will not have a detrimental effect on the substrate. We thank the referee's valuable comment once again!

Overall, the fundamental science in the manuscript is suitable to publish despite some potential shortcomings related to the applications.

Response: We thank the important advice of the referee very much!